# PEAKO and peakTree: Tools for detecting and interpreting peaks in cloud radar Doppler spectra – capabilities and limitations

Teresa Vogl[1,*], Martin Radenz[2,*], Fabiola Ramelli[3], Rosa Gierens[4], and Heike Kalesse-Los[1]

[1]Institute for Meteorology, Leipzig University, Leipzig, Germany
[2]Leibniz Institute for Tropospheric Research, Leipzig, Germany
[3]Institute for Atmospheric and Climate Science, ETH Zürich, Zurich, Switzerland
[4]Institute for Geophysics and Meteorology, University of Cologne, Cologne, Germany
[*]These authors contributed equally to this work.

**Correspondence:** Teresa Vogl (teresa.vogl@uni-leipzig.de)

**Abstract.** Cloud radar Doppler spectra are of particular interest for investigating cloud microphysical processes, such as ice formation, riming and ice multiplication. When hydrometeor types within a cloud radar observation volume have different terminal fall velocities, they can produce individual Doppler spectrum peaks. The peaks of different particle types can overlap, and be further broadened and blended by turbulence and other dynamical effects. If these (sub-)peaks can be separated, properties of the underlying hydrometeor populations can potentially be estimated, such as their fall velocity, number, size and to some extent their shape. However, this task is complex and dependent on the operation settings of the specific cloud radar, as well as atmospheric dynamics and hydrometeor characteristics. As a consequence, there is a need for adjustable tools that are able to detect peaks in cloud radar Doppler spectra to extract the valuable information contained in them. This paper presents the synergistic use of two algorithms used for analyzing the peaks in Doppler spectra, PEAKO and peakTree. PEAKO is a supervised machine learning tool that can be trained to obtain the optimal parameters for detecting peaks in Doppler spectra for specific cloud radar instrument settings. The learned parameters can then be applied by peakTree, which is used to detect, organize and interpret Doppler spectrum peaks.

The application of the improved PEAKO-peakTree toolkit is demonstrated in two case studies. The interpretation is supported by forward simulated cloud radar Doppler spectra by the Passive and Active Microwave TRAnsfer tool (PAMTRA), which are also used to explore the limitations of the algorithm toolkit posed by turbulence and the number of spectral averages chosen in the radar settings.

From the PAMTRA simulations, we can conclude that a minimum number of $n = 20 - 40$ spectral averages is desirable for Doppler spectrum peak discrimination. Furthermore, small liquid peaks can only be reliably separated for eddy dissipation rate values up to approximately $0.0002\,\mathrm{m^2\,s^{-3}}$ in the simulation setup which we tested here.

The first case study demonstrates that the methods work for different radar systems and settings by comparing the results for two cloud radar systems which were operated simultaneously at a site in Punta Arenas, Chile. Detected peaks which can be attributed to liquid droplets agree well between the two systems, as well as with an independent liquid-predicting neural network. The second case study compares PEAKO-peakTree-detected cloud radar Doppler spectra peaks to in situ observations

collected by a balloon-based holographic imager during a campaign in Ny-Ålesund, Svalbard. This case demonstrates the algorithm toolkit's ability to identify different hydrometeor types, but also reveals its limitations posed by strong turbulence and a low $n$.

Despite these challenges, the algorithm toolkit offers a powerful means of extracting comprehensive information from cloud radar observations. In the future, we envision PEAKO-peakTree application on the one hand for interpreting cloud microphysics in case studies. The identification of liquid cloud peaks emerges as a valuable asset e.g. in studies on cloud radiative effects, seeder-feeder processes, or for tracing vertical air motions. Furthermore, the computation of the moments for each sub-peak enables the tracking of hydrometeor populations and the observation of growth processes along fall streaks. On the other hand, PEAKO-peakTree application could be extended to statistical evaluations of longer data sets. Both algorithms are openly available on GitHub, offering accessibility for the scientific community.

# 1  Introduction

Mixed-phase clouds, which contain both ice particles and supercooled liquid water (SLW) at temperatures between 0 and -37°C, are important components of the global climate system because of their large influence on the radiation budget (Sun and Shine, 1994; Tan et al., 2016) and their significant contribution to precipitation, especially over the continents of the northern hemisphere (Mülmenstädt et al., 2015). Yet, despite extensive research on this topic, the understanding of mixed-phase cloud processes is still incomplete due to the complexity of the aerosol-cloud-particle-dynamical system in this type of cloud (Boucher et al., 2013; Korolev et al., 2017). As a result, their representation in numerical weather prediction and climate models remains challenging (Kay et al., 2016; Barrett et al., 2017).

The thorough comparison of high-resolution model simulations with ground-based remote sensing observations is a helpful technique to validate numerical weather models (e.g. Ori et al., 2020; Karrer et al., 2021; Köcher et al., 2023). For this approach, high-quality and high-resolution atmospheric profiling observations need to be available, such as those provided by the platforms of the Aerosol, Clouds and Trace Gases Research Infrastructure (ACTRIS, https://www.actris.eu/) within Cloudnet or the Atmospheric Radiation Measurement (ARM, https://www.arm.gov/) observatories. Vertically pointing Doppler cloud radars are a key component of each of these remote sensing facilities, because of their capability to penetrate thick cloud systems and to yield valuable information about the hydrometeors present at different range levels (Kollias et al., 2007, 2016; Kalesse-Los et al., 2022; Schimmel et al., 2022; Vogl et al., 2022a). For investigating cloud microphysical processes such as riming or secondary ice production (SIP), the cloud radar Doppler spectra are particularly valuable. Cloud radar Doppler spectra represent the backscattered power as a function of Doppler velocity, which, in vertically pointing radars, corresponds to the vertical velocity of particles within the respective observation volume (Kollias et al., 2007). The vertical velocity is determined by the particle fall velocity, which is a function of particle size, shape and density, and atmospheric dynamics. (Zhu et al., 2023). When several populations of hydrometeors with sufficiently different fall speeds (e.g. ice and liquid water) are present in the same observation volume, they each produce a separate peak in the resulting Doppler spectrum. These particle populations can either have different thermodynamic phases, i.e., liquid and ice, different shapes, or even a multi-modal size

distribution. If the peaks can be separated, the moments of each sub-peak can be deduced, which potentially opens possibilities to infer properties of the underlying hydrometeor populations. The separation of Doppler peaks is challenging, because

the peaks of different hydrometeor types can overlap, and turbulence and other dynamical effects further smear the observed Doppler spectrum. Moreover, for large hydrometeors, Mie oscillation of the backscattered power can cause additional minima and maxima in the Doppler spectra, which is especially important at W-band (Tridon et al., 2017).

Sub-peak analysis of cloud radar Doppler spectra can be a powerful tool to interpret processes observed in mixed-phase clouds (e.g. Shupe et al., 2004; Kalesse et al., 2016, 2019; Billault-Roux et al., 2023). The information content is even more enhanced

if spectral dual-polarization or multi-frequency radar measurements are available, which give further insights on the hydrometeor shape and size (e.g. Kneifel et al., 2016; Tridon et al., 2017; Oue et al., 2018; Luke et al., 2021; von Terzi et al., 2022).

Due to recent advances in data storage capacities, the cloud radar Doppler spectra are now usually saved operationally by modern observation facilities. However, despite their valuable information content, they are seldom part of the routine data processing but are often only analyzed in depth for selected case studies. One reason for this is that cloud radar Doppler spectra

are complex and can be difficult to interpret. As described earlier in the text, cloud dynamical features are entangled with microphysical signatures, and all is superimposed with the system noise, resulting in challenges for the interpretation. Furthermore, the use of different operating and system parameters (such as different range and Doppler resolutions) make it very difficult to develop generally valid data analysis techniques. In other words, there are no universally applicable parameters for Doppler spectra peak separation that work for all cloud radars, due to their different resolutions and noise properties.

This highlights the need for adjustable tools that are able to detect peaks in cloud radar Doppler spectra to extract the valuable information contained in them. The tools should ideally integrate into data processing infrastructures such as ACTRIS, be able to rapidly process large amounts of data, and yield intuitive products that can be used to investigate cloud microphysical processes in depth.

Here, we present recent advances in the development of two cloud radar Doppler spectra peak analysis algorithms, PEAKO

(Kalesse et al., 2019) and peakTree (Radenz et al., 2019). PEAKO is a supervised algorithm that is trained on peaks marked in cloud radar Doppler spectra by a human expert for finding the optimal peak detection parameters for specific radar settings. peakTree is an algorithm to identify, structure and interpret peaks in cloud radar Doppler spectra, representing the subpeaks as nodes in a binary tree. Further developments have been made to both tools so that they can now be used in combination: First PEAKO is used for identifying the optimal peak detection settings, and subsequently peakTree is applied for processing the

radar spectra files with the settings found by PEAKO, and saving properties of the detected (sub-)peaks. In this work, we show the potential of the combined cloud radar Doppler spectra peak detection toolkit, in the following referred to as the PEAKO-peakTree toolkit, to derive valuable information about the hydrometeors present in mixed-phase cloud systems. Furthermore, we assess the limitations posed by the employed radar settings and turbulence on the Doppler spectra peak detection.

The paper is structured as follows. In section 2, the data sets used in this study are introduced, followed by a description of

the PEAKO and peakTree algorithms, including a summary of the improvements that have been made since the methods were first published in 2019. In section 3, we present prerequisites for the separability of peaks in cloud radar Doppler spectra based on radar forward simulations. In section 4, the training phase of PEAKO is evaluated and the peak detection parameters found

for the different radar settings are presented. Finally, two case studies demonstrating the application of the PEAKO-peakTree toolkit are presented: In section 4.2, the toolkit is applied to observations from two cloud radar systems, which were operated at the same site in Punta Arenas. For the selected case study, liquid water layers detected by PEAKO and peakTree are validated with lidar observations and an additional independent liquid detection algorithm (Schimmel et al., 2022). In section 4.3, a second case study is presented, in which PEAKO-peakTree results are put into context using in situ observations collected by a balloon-based holographic imager in Ny-Ålesund, Svalbard.

## 2 Methods

### 2.1 Data sets

We are using data from the "Dynamics, Aerosol, Cloud And Precipitation Observations in the Pristine Environment of the Southern Ocean" (DACAPO-PESO, Radenz et al. (2021); Vogl et al. (2022b)) campaign and the Ny-Ålesund AeroSol Cloud ExperimeNT (NASCENT, Pasquier et al. (2022a)) to apply and evaluate the PEAKO-peakTree toolkit. The DACAPO-PESO data set presents a convenient scenario for evaluation, because it includes measurements from two different cloud radar systems operated simultaneously at the same location. Additionally, the NASCENT data set provides an opportunity for PEAKO-peakTree validation, as it contains ground-based cloud radar measurements alongside in situ observations from a holographic imager deployed within the cloud. This combination offers the possibility for direct comparison between the peaks identified in the cloud radar Doppler spectra and the hydrometeor types observed by the in situ instrument. In this section, both data sets, including the instrumentation relevant for this study, are briefly introduced.

### 2.1.1 DACAPO-PESO data set

The DACAPO-PESO experiment aimed at observing aerosol-cloud precipitation processes in the Southern Ocean region, and was carried out from November 2018 to November 2021. In the frame of this project, an extensive suite of remote sensing and in situ instruments was deployed in Punta Arenas, Chile (53.1°S, 70.9°W, Radenz et al., 2021). An overview of the DACAPO-PESO campaign and its goals, the instrumentation and a summary of first results can be found in Vogl et al. (2022b).

Here, we use data observed on 13 March 2019 by two vertically pointing Doppler cloud radars, a pulsed, slanted linear depolarization (SLDR) mode MIRA-35 Ka-band radar (35 GHz, Metek, Görsdorf et al. (2015)) and a dual-polarization frequency modulated continuous wave (FMCW-DP) W-band radar (94 GHz, RPG, Küchler et al. (2017)), referred to in the following text as "LIMRAD94". The two radar systems were operated at the site in close vicinity ($\leq 5$ m distance) for the time period from November 2018 to September 2019. Every hour at 30 minutes past the full hour, MIRA-35 performed a scanning routine, which results in measurement gaps in the vertically pointing observations used in this study. Post-deployment receiver calibration of the MIRA-35 radar revealed a 5 dB low bias, which is why spectra were offset-corrected by $+5$ dB in order to match the reflectivities observed by LIMRAD94 in the Rayleigh regime. LIMRAD94 was operated using three chirps to observe the atmospheric column up to 12 km altitude above the radar. With the aim to ensure good comparability between the instruments,

LIMRAD94 settings were chosen to approximately resemble the vertical and temporal resolution of MIRA-35. More specific
details on the operation settings of the two radar systems and each of the chirps are listed in Table 1, and can be found in
Schimmel et al. (2022). For validation of liquid peaks identified by the PEAKO-peakTree toolkit, we also use data from a
multi-wavelength Raman polarisation lidar of the type Polly$^{XT}$ (Engelmann et al., 2016), which was employed at the site and
measures particle backscatter coefficients at 355, 532, and 1064 nm and extinction coefficients at 355 and 532 nm. Liquid water
path (LWP) observations by a RPG HATPRO G2, a 14-channel microwave radiometer (MWR, Rose et al., 2005), are also used
in this study.

### 2.1.2 NASCENT campaign and long-term observations at AWIPEV

NASCENT took place at the French-German Arctic Research Base AWIPEV in Ny-Ålesund, Svalbard (78.9°N, 11.9°E) from
September 2019 to August 2020. It was targeted to study the microphysical and chemical properties of aerosols and clouds in
the Arctic using an extensive set of in situ and remote sensing instrumentation. Here, we show the PEAKO-peakTree applica-
tion on a case observed on 12 November 2019, which was previously analyzed in detail by Pasquier et al. (2022a). A detailed
description of the NASCENT campaign, the field sites and the instrumentation can also be found in Pasquier et al. (2022a).
In situ cloud microphysical measurements were obtained with the tethered balloon system HoloBalloon (Ramelli et al., 2020)
up to an altitude of 1000 m above ground (Pasquier et al., 2022a, b). The main instrument on the measurement platform was
the HOLographic Imager for Microscopic Objects (HOLIMO), which was mounted 12 m below the balloon. HOLIMO uses
digital in-line holography to image an ensemble of cloud particles in the size range from 6 $\mu$m to 2 mm in a three-dimensional
sample volume (Henneberger et al., 2013; Ramelli et al., 2020). Based on the captured particle images, the phase-resolved size
distributions and shapes of ice crystals and cloud droplets can be obtained. The classification into cloud droplets and ice crys-
tals is done using a convolutional neural network based on the particle shape (Touloupas et al., 2020). The phase partitioning
between cloud droplets and ice crystals is only done for particles larger than 25 $\mu$m (i.e., all particles below this size threshold
are classified as cloud droplets), because the resolution of HOLIMO is not sufficient to distinguish the shape of smaller par-
ticles (Henneberger et al., 2013). In addition, all ice crystals were manually classified into the following ice habits using the
particle shape information: columns, plates, frozen drops, irregular ice crystals and aged particles (i.e., rimed and aggregated
ice crystals). A sample volume of 15.5 cm$^3$ and a frame rate of 6 frames per second were considered for the analysis of the
HOLIMO data, resulting in a total sample volume rate of 5.6 L min$^{-1}$ (Pasquier et al., 2022a).
Continuous W-band cloud radar measurements have been performed at AWIPEV since 2016 (Nomokonova et al., 2019;
Chellini et al., 2023). During the time period considered in this study, a single-polarization FMCW radar (94 GHz, RPG,
(Küchler et al., 2017)) was employed at the site, referred to in the following as "JOYRAD94". The measurements were set up
with high vertical ($< 10$ m in the lowest three chirps) and temporal ($< 2$ s) resolution. However, as a consequence, the Nyquist
velocity was rather low, ranging between 5.1 and 3.2 m s$^{-1}$, and the number of spectral averages $n$ was very low, between 4
and 8 for the four chirps (see Table 1). In this work, the level 0 data provided by the instrument software were used, containing
the raw cloud radar reflectivity Doppler spectra. For PEAKO and peakTree, these files were processed in an additional step,
which removes the signal below the noise threshold, which is defined as the mean noise + 6 standard deviations of the noise

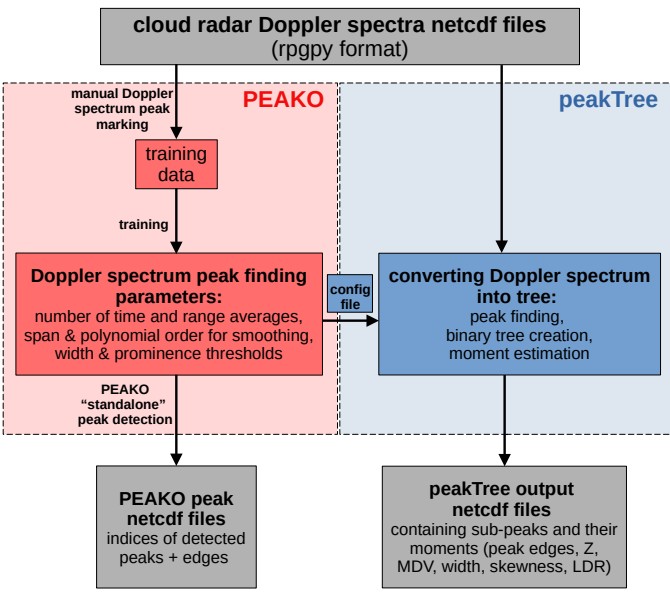

**Figure 1.** Flowchart summarizing the PEAKO and peakTree algorithm work flow, along with input/ output files. Cloud radar Doppler spectra in the rpgpy netcdf format are required by PEAKO and one option for peakTree, which also includes functions to read in Doppler spectra in other formats.

(Hildebrand and Sekhon, 1974). No additional aliasing correction has been performed, as this effect did not play a role for the selected case study. Every 1000 s, an internal calibration cycle interrupts the measurements, causing a gap of about 40 s.

## 2.2 PEAKO and peakTree algorithm descriptions and updates

In the following section, PEAKO and peakTree are introduced and updates made to the code since 2019 are summarized. Fig. 1 shows an overview of the work flow, the input and output data and the interface between the two algorithms. While both of them can be used standalone, this work focuses on the added value of using PEAKO to derive the peak detection parameters which are then transferred to peakTree via a configuration file and applied for peak detection.

### 2.2.1 PEAKO

PEAKO (Kalesse et al., 2019) is a supervised algorithm for peak detection in cloud radar Doppler spectra. While the original software version was written in Matlab, the current PEAKO implementation is in Python. Cloud radar Doppler spectra files are expected in a format as yielded by the netcdf generation module of the rpgpy Python package (https://github.com/actris-cloudnet/rpgpy). In the first step, the training data generation (Fig. 1), training data containing hand-marked peaks in cloud radar Doppler spectra are generated by a human expert labeller. When choosing training data, care should be taken to

ensure that the type of (meteorological) targets to be studied later with peakTree are adequately represented in the data set. For example, if liquid water peaks are the focus of the peakTree study, it should be ensured that liquid-containing spectra are included in the training data set. Furthermore, it may be practical to use classification algorithms such as the Cloudnet target classification (Illingworth et al., 2007; Tukiainen et al., 2020), to include only data classified as "cloud", "rain" or "insects" in the training data set, depending on the envisioned application.

PEAKO contains a peak detection function with a set of adjustable parameters, which are in the second step, i.e. in the algorithm training phase, tuned to achieve the best match with the training data set. In the original algorithm description published in Kalesse et al. (2019), the peak detection function took an averaged Doppler spectrum (i.e. averaged over a fixed window in time and range domains) as input. The number of time and range averages have now been added to the set of adjustable parameters in PEAKO and is varied during the training phase.

Following time- and range averaging, the spectrum is smoothed. The smoothing algorithm, a Savitzky-Golay filter, takes a span (window size) as input, which is one of the adjustable PEAKO parameters. This span is now defined in $\mathrm{m\,s^{-1}}$ as opposed to the fraction of the total number of data points (Doppler bins) used in the first PEAKO version. Over this window size, a polynomial is fit to the observational data. The degree of the polynomial (originally fixed to 2) is now an adjustable PEAKO parameter as well, which can be varied during the training phase.

Finally, peak detection is performed on the averaged and smoothed Doppler spectrum. Only peaks with widths larger than the width threshold and prominences above the prominence threshold are considered and saved. The width is defined as the peak width at half-height. The peak prominence is the power difference between the maximum value of the considered peak and the minimum between this peak and the closest higher neighboring peak.

This results in six adjustable PEAKO parameters: The number of averaged spectra in the time and range domains, respectively, the span and the polynomial order used by the smoothing module, and the width and prominence thresholds applied by the peak detection function. PEAKO is trained by looping over all possible parameter combinations in a defined search grid and by computing a similarity measure based on the overlapping area below human-marked peaks and those yielded by the peak detection function. The parameters leading to the highest similarity score are saved and can be used as input for peakTree via a configuration file (Fig. 1).

In the original algorithm description, the learning process was divided into two phases, a training and a test phase. In the current PEAKO version, the training phase can be set up to include $k$-fold cross-validation, where the training data set is split into $k$ subsets ("folds"). Training is then performed $k$ times on $k-1$ folds, whereas each time another one of the $k$ folds is used for validation. The test data set, which was not seen by PEAKO during the training/ validation phase, is then used for final evaluation of the learned parameters. PEAKO can be used as a standalone tool (Kalesse et al., 2019; Billault-Roux et al., 2023) to detect and save the indices of peaks in cloud radar Doppler spectrum files. However, for more in-depth analyses and fast processing of larger data sets, the use of peakTree is recommended.

### 2.2.2 peakTree

The peakTree technique introduced in Radenz et al. (2019) uses a binary tree structure to recursively represent peaks and subpeaks in a Doppler spectrum. An element inside the tree structure is called node, where each node holds the spectral moments of a part of the Doppler spectrum identified by the peak finding algorithm. The root node contains the complete signal of the Doppler spectrum above the noise threshold. Starting from the noise floor, the peaks are recursively split into two so-called children, if they contain subpeaks (identified by local minima in spectral reflectivity). One advantage of using peakTree instead of the PEAKO standalone peak detection functionality is that, for each node, the moments of the spectrum are calculated: The reflectivity Z, mean Doppler velocity MDV, spectrum width, skewness, and the linear depolarization ratio LDR for dual-polarization radars. Please note that, throughout this paper, negative MDV values represent downward velocities in the vertical column (i.e. towards the radar). Another advantage is the use of Numba (Lam et al., 2015), which means a considerable improvement in terms of computation time compared to PEAKO. Recently, the option for reading in the RPG FMCW binary format spectra files via rpgpy was added. Furthermore, the peak finding algorithm was adapted to be compatible with the training output of PEAKO. This means that the internal peak detection function in peakTree now resembles the one used by PEAKO, and that PEAKO training parameters (averaging in time and range, the span and polynomial order used for smoothing, and minimum peak width and prominence) can be directly supplied to peakTree via a configuration file (Fig. 1). It should be noted that peakTree can also be used without PEAKO's optimized peak detection parameters (Radenz et al., 2019; Ramelli et al., 2021). In this case, the peak detection parameters have to be chosen manually in a rather subjective try-and-error procedure. We thus suggest the combined use of PEAKO-peakTree.

peakTree offers the possibility to detect certain hydrometeor types corresponding to subpeaks in the cloud radar Doppler spectra. Nodes resembling characteristic particle populations can be selected by straightforward, threshold-based rules: Cloud droplets usually have negligible terminal velocity and a low reflectivity, due to their small size. Hence, a very basic "liquid node" selection rule based on two thresholds already gives useful results:

$$Z_{liquid} < -20\,\mathrm{dBZ} \tag{1}$$

$$|MDV_{liquid}| < 0.3\,\mathrm{m\,s^{-1}} \tag{2}$$

It should be noted that liquid peaks in up- or downdrafts larger than $0.3\,\mathrm{m\,s^{-1}}$ are not detected by this simple selection rule and that small ice can result in peaks with similar properties. For dual-pol radars, an additional criterion can be added to filter peaks with elevated LDR, which are characteristic for columnar ice particles.

Furthermore, it is often of interest to cloud microphysical studies to consider hydrometeors with large fall velocities, which, in the temperature range for mixed-phase clouds between 0 and -40°C, usually correspond to large frozen drops and strongly rimed particles. Here, this "fast-falling" population is identified by a slightly more complex node selection rule, where the node index is 1 or 3 (i.e., the fastest falling subpeak of the spectrum), and the following thresholds:

$$MDV_{fast} < -1.8\,\mathrm{m\,s^{-1}} \tag{3}$$

$$prominence_{fast} > 2\,\mathrm{dBZ} \tag{4}$$

In this respect, peakTree can in principle also be used to filter ground clutter, which usually emerges as Doppler spectrum peaks centered at $0\,\mathrm{m\,s^{-1}}$ (Williams et al., 2018). However, in both data sets used in this study, clutter did not play a role, i.e. a removal of peaks due to non-meteorological targets was not required.

## 2.3 VOODOO

We are using an independent liquid detection method to validate the liquid-containing Doppler spectrum peaks detected by the PEAKO-peakTree toolkit. The VOODOO (reVealing supercOOled liquiD beyOnd lidar attenuatiOn) retrieval (Schimmel et al., 2022) uses deep convolutional neural networks to yield a probability of the presence of cloud droplets (CD) in a given radar observation volume. It uses cloud radar Doppler spectra as input and was trained using the Cloudnet target classification as supervisor. Its ability to accurately predict the presence of liquid water was demonstrated in the DACAPO-PESO data set by
Schimmel et al. (2022). Here, we use the VOODOO-predicted probability of cloud droplets as validation for the liquid water peak detection by PEAKO and peakTree.

## 2.4 Eddy dissipation rate retrieval

We are using the eddy dissipation rate (EDR, $\epsilon$) as a measure of atmospheric turbulence to identify regions in the cloud where turbulence potentially affects the detectability of sub-peaks in cloud radar Doppler spectra. The EDR is the rate at
250 which turbulent kinetic energy cascades in the atmosphere from large to smaller and smaller eddies, until it is converted from mechanical into thermal energy at the molecular level (Foken, 2008). It is a universal measure of turbulence, i.e. a large EDR means that the atmosphere is dissipating energy quickly, and that the atmospheric turbulence level is high. EDR can be retrieved from cloud radar measurements of MDV as follows (Borque et al., 2016; Griesche et al., 2020): The time series of MDV measured at one radar range gate in a 5-minute time interval is converted to a power spectrum using a Fast Fourier
Transform (FFT). Furthermore, a transition from time to spatial frequencies is done assuming Taylor's hypothesis of frozen (isotropic) turbulence (Taylor, 1938):

$$\nu = f/v_h \tag{5}$$

where f is the frequency and $\nu$ is the wavenumber $(m^{-1})$, which can be related to a characteristic turbulent structure size, or eddy length scale $\lambda$, via

$$\lambda = 2\pi/\nu \tag{6}$$

$v_h$ is the horizontal advection wind speed. To estimate $v_h$, wind data from the European Centre for Medium-Range Weather Forecasts (ECMWF) Integrated Forecast System (IFS) are used for the DACAPO-PESO data and wind data from the icosahedral non-hydrostatic (ICON) model for the NASCENT data set. For retrieving the EDR from the resulting spectrum, the following assumption is made: In the inertial subrange, turbulence is assumed to be isotropic and the decrease in energy is
265 proportional to $\mathrm{f}^{-5/3}$. The power spectrum $S$ of turbulent energy dissipation can thus be represented as:

$$S(\nu) = A\epsilon^{2/3}\nu^{-5/3} \tag{7}$$

With $A = 0.5$ the Kolmogorov constant. Consequently, if the spectrum derived from measured MDV can be expressed in this way, the EDR $\epsilon$ can be inferred. Following Griesche et al. (2020), linear least-square regressions of the spectrum in 34 different wavenumber $\nu$ intervals are performed, yielding intercept $i$ and slope $s$ parameters. If $s$ is within $-5/3 \pm 1/3$, the fit is considered "good" and EDR is calculated according to:

$$\epsilon = \left( \frac{10^i}{A} \right)^{3/2} \tag{8}$$

If the $s = -5/3 \pm 1/3$ criterion is met by more than one of the 34 intervals, $\epsilon$ is calculated using $i$ from all of these intervals and the mean $\epsilon$ is used.

## 2.5 PAMTRA

The Passive and Active Microwave TRAnsfer (PAMTRA) model (Mech et al., 2020) is used for forward-simulating cloud radar Doppler spectra using in situ observations of liquid and ice particles collected by HOLIMO following Pasquier et al. (2022a). Based on the particles detected by HOLIMO, the following particle populations are used in radar forward simulations: liquid, (small) columnar ice, rimed aggregates and frozen droplets.

The liquid particle number size distribution (PNSD) is frequently bimodal, and we split it into a cloud and a drizzle population at a diameter of $60 \, \mu$m. We are using HOLIMO in situ observations with $60 \, s$ time resolution. This decision results from a trade-off between better matching the radar's time resolution, and increasing the volume observed by HOLIMO by averaging over longer time periods. Here, we choose the $60 \, s$ time resolution to keep the relatively high temporal resolution. In addition, for each hydrometeor type in the HOLIMO in situ data, bins without counts within the size distribution (i.e., between the smallest and largest particles detected) are filled with a value of $10 \, m^{-3}$ to reduce the problem of artificial sub-peaks in the forward-simulated spectra, caused by insufficient counting statistics.

Terminal velocity-size relations for the non-spherical particles, i.e. columns and rimed aggregates, follow Heymsfield and Westbrook (2010), while cloud, drizzle and frozen drops follow Khvorostyanov and Curry (2002). Mass-size and cross section area-size power law relationships for the ice columns are taken from Mitchell (1996). For the rimed aggregates, these relationship coefficients are taken from Maherndl et al. (2023), who presented these parametrizations for aggregates of different monomers, depending on the normalized rime mass $M$. $M$ is defined following Seifert et al. (2019) as:

$$M = \frac{m_{rime}}{m_g} \tag{9}$$

i.e. the ratio of the rime mass $m_{rime}$ and the mass of a size-equivalent spherical graupel $m_g$. For our simulations, we initially varied $M$ between 0.2 and 0.8 and finally picked a value of 0.2, matching the fall speed of the observed radar Doppler spectra best.

In our PAMTRA simulations, the vertical air velocity is set to $0 \, \mathrm{m \, s^{-1}}$. Turbulence broadening can be included via the kinematic broadening of Doppler spectra $\sigma$:

$$\sigma = \sqrt{\sigma_B^2 + \sigma_T^2} \tag{10}$$

where $\sigma_B$ is the broadening due to finite beam width, assuming a horizontal wind velocity $v_h$ and the radar beam width $\theta$, and $\sigma_T$ is the broadening due to turbulence within the radar sampling volume, which is calculated via

$$\sigma_T = \sqrt{3 \cdot A/2 \cdot \left(\frac{\epsilon}{2\pi}\right)^{2/3} \cdot (L_s^{2/3} - L_\lambda^{2/3})} \tag{11}$$

$L_s$ and $L_\lambda$ are the maximum and minimum length scale observable by the radar. To obtain the total kinematic broadening $\sigma$, $v_h$ is assumed to be $5.5\,\mathrm{m\,s^{-1}}$, and the radar beamwidth $0.3°$. For example, for the radar settings chosen in our simulations, an EDR of $0.0001\,\mathrm{m^2\,s^{-3}}$ translates to a broadening of $\sigma = 0.05\,\mathrm{m\,s^{-1}}$. A higher EDR of $0.0065\,\mathrm{m^2\,s^{-3}}$ would correspond to $\sigma = 0.19\,\mathrm{m\,s^{-1}}$. In our simulations, we varied $\sigma$ between $0.02$ and $0.2\,\mathrm{m\,s^{-1}}$.

The range resolution, number of FFT points, number of spectral averages $n$ and Nyquist velocities were set in PAMTRA similar to the JOYRAD94 settings (see Table 1). For each Doppler spectrum, multiple simulations were performed to account for the separate contributions of each single particle population to the combined spectrum.

Please note that the forward simulation results are strongly dependent on the employed mass-size relations and particle fall velocity-size relations, especially considering the ice phase. The goal in this study is however not to reach closure between the

forward-modeled and observed radar variables, but to instead explore the effect of radar settings (e.g. the number of spectral averages) and EDR on the forward-simulated radar Doppler spectra. While acknowledging the sensitivity of our simulations on these parametrizations, we do not expect a general influence on the separability of cloud radar Doppler spectrum sub-peaks.

## 3   Conditions for peak separability in cloud radar Doppler spectra

Different particle populations cannot always be distinguished in the observed Doppler spectrum. Even in conditions with ho-

mogeneous radar beam-filling, and when the mean vertical wind is constant over the sampling period, turbulence and vertical wind shear broaden the spectrum to an extent that might blur boundaries between subpeaks caused by different particle populations (Shupe et al., 2008). Moreover, instrument settings not optimized for peak detection in Doppler spectra might result in spectra that are too noisy to detect the subpeaks. If the number of incoherent averages $n$, i.e. the number of spectra averaged after the FFT, is set too low, this results in strong phase noise signatures, which are a challenge for multi-peak analysis (Zrnić,

1975).

To assess the question under which conditions different particle populations can be separated in a radar Doppler spectrum, we are using balloon-borne in situ data from the NASCENT campaign as input in idealized PAMTRA simulations. We chose a one-minute time interval (12 November 2019 16:01 UTC) representative for complex situations with multiple particle populations to forward-simulate cloud radar Doppler spectra from the PNSDs measured by HOLIMO. The spectrum broadening $\sigma$

is varied from $0.02$ to $0.2\,\mathrm{m\,s^{-1}}$, and different radar instrument settings including 6, 10, 20, 40, 80 and 120 incoherent averages, respectively, are considered. For each set, 100 spectra are simulated to account for the random nature of the phase noise. peakTree is used to detect peaks in the forward-simulated cloud radar Doppler spectra, employing peak finding parameters that were optimized by PEAKO for each set of spectra with the respective $n$.

Fig. 2a shows the number of spectral averages vs. the number of detected peaks. The spectra analyzed with the optimized

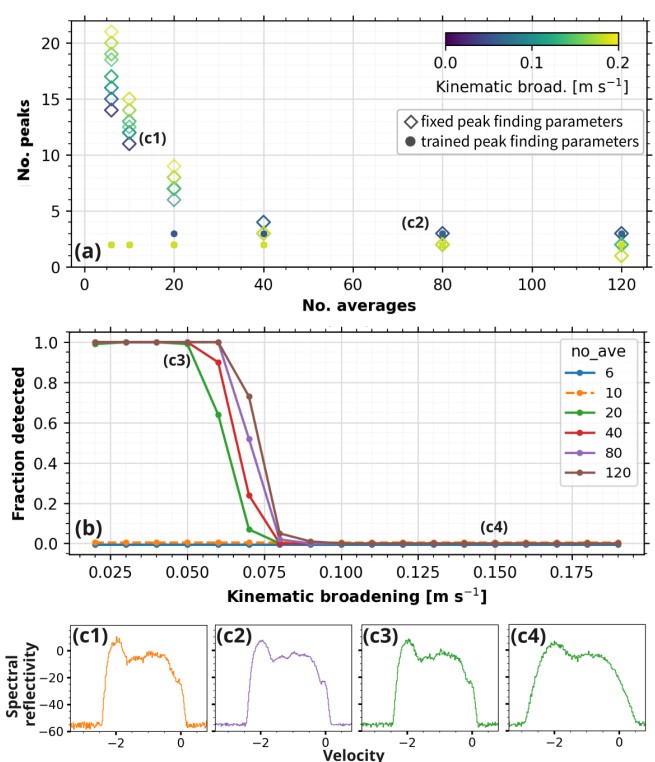

**Figure 2.** PEAKO-peakTree peak detection results in forward-simulated cloud radar Doppler spectra using PAMTRA a) number of detected peaks by PEAKO-peakTree vs. the number of spectral averages $n$ set in the forward simulation. Compared are two sets of peak finding parameters: fixed (indicated by open diamonds; a prominence threshold of $1.5\,\text{dB}$, minimum width of $0.03\,\text{m s}^{-1}$, and no smoothing along the velocity dimension) and optimized by PEAKO individually for each $n$ (dots). The marker color represents the turbulence broadening in $\text{m s}^{-1}$ set in the PAMTRA forward simulation. b) fraction of succesfully detected liquid peaks in 100 forward-simulated spectra vs. turbulence broadening in $\text{m s}^{-1}$ for radar settings with $n = 6$, 10, 40, 80 and 120. c1–c4) examples of forward-simulated spectra: c1 with $n = 10$ and $\sigma = 0.05\,\text{m s}^{-1}$; c2 with $n = 80$ and $\sigma = 0.05\,\text{m s}^{-1}$; c3 with $n = 20$ and $\sigma = 0.05\,\text{m s}^{-1}$; c4 with $n = 20$ and and $\sigma = 1.5\,\text{m s}^{-1}$. The corresponding $\sigma$ and $n$ of c1–4 are also marked in panels a) and b)

peak finding parameters (Fig. 2a dots) show two peaks for the high turbulence conditions and 3 peaks for lower turbulence conditions, where color indicates the turbulence strength. For comparison, the number of peaks is also shown for a set of fixed peak finding parameters (Fig. 2a open diamonds). For a low number of averages $n$, an unrealistically high number of peaks is detected using the fixed parameters, which can be explained by the increased false alarm rate due to noise. This is also visible in the example spectrum in Fig. 2c1 with $n = 10$. Notably, for low number of averages, increased turbulence increases

the number of detected peaks. This can be explained as follows: For noisy spectra (i.e. $n < 20$), broadening due to turbulence increases the number of Doppler bins above the noise floor and hence increases the probability of additional peaks. In contrast, for smooth spectra (i.e. higher $n$), additional turbulence hides small peaks.

The spectrum in Fig. 2c1 represents an example for a noisy spectrum ($n$=10) and low atmospheric turbulence levels ($\sigma = 0.05 \, \mathrm{m \, s^{-1}}$). The same spectrum is forward-simulated with $n$=80 in Fig. 2c2. Three out of five peaks can be identified by

peakTree here, corresponding to the hydrometeor populations with different terminal fall velocities. The effect of turbulence broadening is illustrated in the spectra in Fig. 2c3 and c4, where $n$=20 and $\sigma = 0.05$ and $0.15 \, \mathrm{m \, s^{-1}}$, respectively. In Fig. 2c4, only two broad peaks are visible in the forward-simulated spectrum.

These combined effects of noise and turbulence become more tangible if we consider the question whether the liquid peak can be detected or not. For this purpose, the peakTree liquid peak selection rule was applied to the detected nodes (Eq. 1 - 2). A

liquid peak is only considered to be detected successfully if its reflectivity and MDV deviate less than $1 \, \mathrm{dB}$ and $0.1 \, \mathrm{m \, s^{-1}}$ from the reference obtained for 150 coherent averages and weak turbulent broadening. Fig 2b shows the fraction of detected liquid peaks as a function of turbulence broadening $\sigma$. For spectra with a low number of averages, the liquid peak was not found regardless the turbulence broadening. When increasing the number of averages to $n > 10$, the liquid peak is detected almost in all cases in conditions with low spectrum broadening (i.e. low turbulence). At approximately $\sigma = 0.06 \, \mathrm{m \, s^{-1}}$, the fraction

of successfully detected liquid peaks declines strongly for all simulations. This $\sigma$ value corresponds approximately to an EDR of $0.0002 \, \mathrm{m^2 \, s^{-3}}$, i.e. a medium level of turbulence, which can be observed in almost all types of clouds (Shupe et al., 2012; Borque et al., 2016; Griesche et al., 2020).

From our simulations, we can conclude that a) a minimum number of 20 - 40 spectral averages is desirable in radar settings optimized for peak detection, and that b) liquid peaks can only be separated for EDR values up to approximately $0.0002 \, \mathrm{m^2 \, s^{-3}}$

(translating to a $\sigma$ of approximately $0.06 \, \mathrm{m \, s^{-1}}$) under such complex multi-peak situations. Appendix A shows that for simpler conditions (PNSDs featuring a much stronger liquid peak), a separation may however still be possible even for low values of $n$ and slightly higher values of turbulence broadening.

## 4   Results

In this section, PEAKO training results are presented (section 4.1), followed by two case studies of mixed-phase clouds in

sections 4.2 and 4.3, where the application of the PEAKO-peakTree toolkit is demonstrated. In the first case study (section 4.2) from the DACAPO-PESO data set, the liquid-peak-finding results are validated with the independent cloud liquid detection method VOODOO. In the second case study from the NASCENT campaign (section 4.3), the detected Doppler spectrum peaks

are set into context with PAMTRA forward simulations of cloud radar Doppler spectra based on PNSDs of ice and liquid observed with HOLIMO.

## 4.1 PEAKO training results

To answer the question how many cloud radar Doppler spectra are needed for the training result to converge, PEAKO training runs with different numbers of training samples were performed. As the different chirps of the FMCW radars have different radar settings, each of the chirps requires individual PEAKO peak finding parameters. This results in a total of eight different sets of radar settings for the two data sets considered in this study: LIMRAD94 was employed with three chirps, MIRA-35 is a pulsed radar that has the same settings throughout the observational range, and JOYRAD94 was operated with four chirps during NASCENT. For each set of radar settings, five different training runs with up to 500 hand-marked spectra were performed, meaning that for each radar configuration, peaks were marked in more than 2,500 cloud radar Doppler spectra by a human expert. For the training data generation, observations that do not overlap with the case studies shown in Sections 4.2 and 4.3 were selected. For LIMRAD94, observations from seven hours, when mixed-phase cloud systems were present in Punta Arenas during DACAPO-PESO, were used. In addition, Doppler spectra from four hours of observations obtained at Leipzig University, measured with the exact same radar settings as during DACAPO-PESO, were selected. MIRA-35 training data include observations from four hourly Doppler spectra files obtained during DACAPO-PESO. For JOYRAD94, training data were generated using six hourly Doppler spectrum files from 11 November 2019, 12 November 2019 and 4 April 2020.

Fig. 3 shows the number of training samples versus the PEAKO similarity score reached during the training and validation runs in the $k$-fold cross-validation for the different cloud radars and settings. The PEAKO similarity score is based on the overlapping area of algorithm-detected and human-marked peaks (see section 2.2). For low training sample sizes ($\lesssim 100$), the validation score is lower than the training score, whereas the scores converge with increasing sample size (Fig. 3). The high training score for small training data set sizes is due to overfitting, where the algorithm is trained to fit a small number of training data very well. The same model fails however, when it is applied to the validation data set, which was not used during training. Thus, the minimum number of cloud radar Doppler spectra needed to train PEAKO optimally is where the training and validation lines converge. Fig. 3 shows that, while this point is reached at different numbers for the different radar settings, generally 150 to 200 spectra seem to be a sufficient number of hand-marked cloud radar Doppler spectra to train PEAKO. Differences in the maximum achievable similarity score in the training/validation curves for the different radar settings can be explained by differences in the radar settings and hydrometeors observed. When the lines converge at a value lower than 100%, this means that the peak detection function is unable to match (only) all human-marked peaks in the training data set. The reflectivities of undetected and falsely detected peaks are subtracted from the PEAKO similarity score. Instrument artifacts, such as "ghost echoes" and Doppler spectrum "tails", which are not completely filtered from the data set before the training phase, cause spurious peaks that are easily recognized when manually marking peaks, but are impossible to distinguish for the automatic algorithm. This explains why the maximum achievable skill score is below 100% for LIMRAD94 and MIRA-35. The training and validation scores for the first three chirps of JOYRAD94 (Fig. 3b,d,f) also converge at values distinctly below 100%, although artifacts are less frequently present in the NASCENT data set. However, JOYRAD94 was operated with only

**Table 1.** Overview of used radar settings (upper half) and PEAKO training results (lower half).

| | LIMRAD94 | | | MIRA-35 | JOYRAD94 | | | |
| --- | --- | --- | --- | --- | --- | --- | --- | --- |
| | chirp 1 | chirp 2 | chirp 3 | (pulsed) | chirp 1 | chirp 2 | chirp 3 | chirp 4 |
| **range resolution (m)** | 29.81 | 44.72 | 39.75 | 31 | 3.20 | 7.45 | 9.67 | 23.85 |
| **range extent (km)** | 0.12 – 1.19 | 1.21 – 6.98 | 7.04 – 11.96 | 0.16 – 15.65 | 0.1 – 0.4 | 0.4 – 1.19 | 1.21 – 3 | 3.01 – 12 |
| **number of averaged spectra $n$** | 23 | 54 | 124 | 20 | 8 | 6 | 4 | 6 |
| **integration time (s)** | 0.72 | 0.62 | 1.42 | 2.0 | 0.37 | 0.27 | 0.37 | 0.27 |
| **number of FFT points** | 256 | 256 | 128 | 512 | 512 | 512 | 512 | 256 |
| **Nyquist velocity (m s$^{-1}$)** | 9.0 | 6.3 | 4.7 | 10.5 | 5.1 | 5.1 | 3.2 | 3.2 |
| **Doppler resolution (m s$^{-1}$)** | 0.07 | 0.049 | 0.073 | 0.041 | 0.02 | 0.02 | 0.013 | 0.025 |
| **number of averaged time steps** | 0 | 3 | 0 | 3 | 0 | 3 | 3 | 0 |
| **number of averaged range bins** | 0 | 3 | 0 | 0 | 0 | 3 | 3 | 0 |
| **span for smoothing (m s$^{-1}$)** | 0.25 | 0.2 | 0.1 | 0.2 | 0.15 | 0.2 | 0.2 | 0.15 |
| **polynomial order for smoothing** | 2 | 2 | 2 | 2 | 1 | 3 | 3 | 2 |
| **min. peak width (m s$^{-1}$)** | 0 | 0 | 0 | 0 | 0 | 0 | 0 | 0 |
| **min. peak prominence (dB)** | 0.5 | 0.5 | 1.5 | 1.0 | 2.0 | 2.0 | 4.0 | 2.0 |

$n = 4$–$8$ averaged spectra per sample (see Table 1), resulting in very noisy spectra, presenting a challenge to the human expert labellers trying to identify peaks in these radar Doppler spectra. Moreover, the range covered by the lowest three JOYRAD94 chirps in the vertical column extends from 0 to 3 km. In this region, multiple hydrometeor populations with different fall speeds occur frequently, leading to multi-modal cloud radar Doppler spectra. The training data set thus contains very noisy, multi-modal radar Doppler spectra, where the labeller was uncertain whether to mark a peak or not, making it impossible for a peak detection function to match all the human-marked peaks.

In contrast, the training and validation scores in Fig. 3e and h converge at high scores >80 %. These are chirp 3 of LIMRAD94 and chirp 4 of JOYRAD94, i.e. the highest chirps in the vertical column starting at 7 and 3 km, respectively. The training spectra in these chirps are almost exclusively monomodal, facilitating the peak detection in the Doppler spectra.

## 4.2 Case study 1: Multi-liquid cloud layers observed during DACAPO-PESO

In this case study, peakTree is applied to Doppler spectra data from the W-band and Ka-band cloud radars deployed during DACAPO-PESO using the optimized peak detection parameters yielded by PEAKO. The reflectivity of the liquid cloud peak is derived by peakTree using the thresholds defined in Eq. 1 and 2. The purpose of this comparison is twofold: Firstly, to verify that the peaks derived for the two individual radar systems match, which serves as a validation of the proposed PEAKO-peakTree methodology. Secondly, this case demonstrates the capability of PEAKO-peakTree to identify (supercooled) liquid water layers in mixed-phase clouds, which is of considerable value for various research applications.

For the aforementioned comparison, we choose to analyze a case where multiple liquid-containing cloud layers were observed

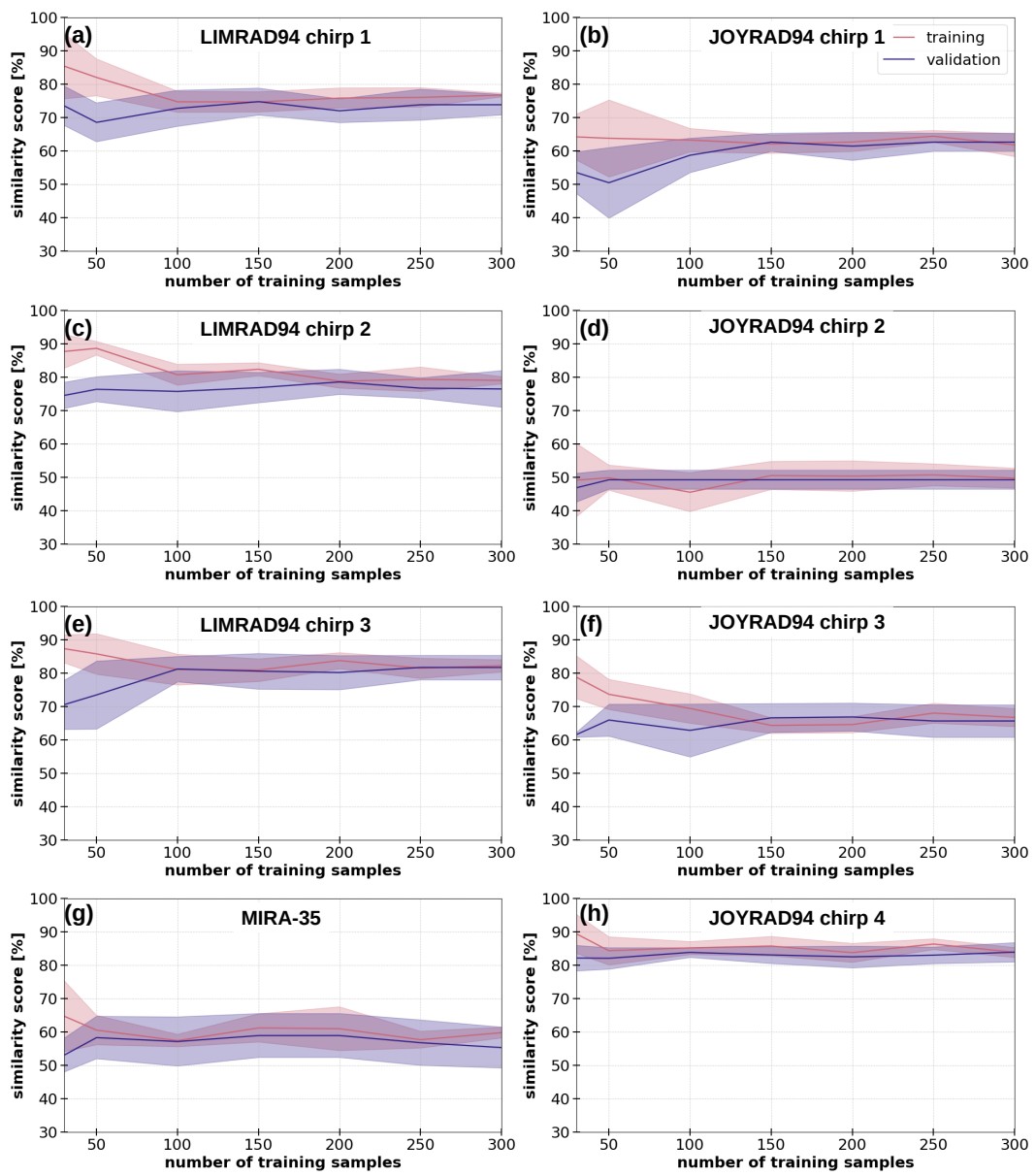

**Figure 3.** Number of training samples vs. PEAKO similarity score for the training (red) and validation (blue) runs. The bold lines represent the mean of five training/ validation runs, and the shaded area ± one standard deviation.

on 13 March 2019 in Punta Arenas. On this particular day, there was a notable shift from the prevailing westerly winds to easterly winds, coinciding with the presence of a multilayer cloud system with cloud tops between 4 and 7 km altitude (Fig. 4a). The melting layer was located at approximately 1.5 km altitude. Weak precipitation ($\leq 1\,\mathrm{mm\,h^{-1}}$) was observed at around 11:30 UTC. The liquid water path measured by the MWR ranged between approximately 50 and $200\,\mathrm{g\,m^{-2}}$ for the considered time period from 8 to 12 UTC.

The MDV observed by LIMRAD94 (Fig. 4b) reveals several horizontal layers exhibiting zero or positive MDV, thereby suggesting the presence of multiple embedded liquid water-containing layers. Some of these layers are also discernible in the attenuated lidar backscatter observed by Polly$^{XT}$ (Fig. 4d). Because the lidar signal is strongly attenuated by liquid water, it is often unable to penetrate thick cloud systems or multi-layered clouds containing multiple liquid layers. However, between approximately 10 and 11:30 UTC, up to four liquid layers are visible.

VOODOO offers a practical option to extend the range up to which liquid can be observed beyond the point where the lidar signal is completely attenuated by liquid water. In Fig. 4c, the VOODOO-predicted probability of liquid cloud droplets, $p_{CD}$, is shown. Yellow-ish layers indicate a high probability for the presence of cloud droplets in the respective observation volume. For better comparability of the VOODOO-predicted $p_{CD}$ and the attenuated backscatter coefficient observed by the lidar, pixels with $p_{CD} > 0.65$ are overlaid on Fig. 4d. In this visualization, it becomes apparent that these regions with elevated $p_{CD}$ match well with the areas exhibiting high attenuated backscatter coefficient in those parts of the cloud, which can be penetrated by the lidar. In higher cloud layers (beyond 3 km), VOODOO liquid-containing layers are noticeably thicker than those observed by the lidar, with the layer of elevated lidar backscatter being located at the lower boundary of the pink structure representing high $p_{CD}$ regions. This suggests that the geometric extent of liquid water layers is larger than observed by the lidar, because its signal becomes completely attenuated at some point in the cloud.

When using VOODOO, one limitation needs to be considered however, i.e. that it is solely based on the cloud radar reflectivity Doppler spectra, leading to the potential misclassification of small and narrow spectral peaks caused by ice as liquid. In regions immediately beneath layers of SLW, ice crystals newly formed through primary or secondary ice formation processes can be present. Due to their small diameters and consequently low fall velocities, these pristine ice crystals produce peaks in the cloud radar Doppler spectra that closely resemble those of liquid cloud droplets.

The number of peaks detected by PEAKO in LIMRAD94 and MIRA-35 Doppler spectra are shown in Fig. 4e and f, respectively. For each (sub-)peak, the moments are calculated and hydrometeor classification thresholds, as described in Section 2.2.2, are applied. The two panels in Fig. 4g and h show the reflectivity of the liquid peak yielded by the PEAKO-peakTree toolkit for LIMRAD94 and MIRA-35, respectively. Remarkably, the liquid layers detected by PEAKO and peakTree for the two different cloud radar systems are very similar and also align closely with $p_{CD}$ estimated by VOODOO. Compared to lidar-detected liquid layers (Fig 4d), further layers near the cloud tops with temperatures around -20°C (8:30 – 9:00 UTC) and -12°C (8:30 – 10:00 UTC), pointing to the existence of SLW-containing layers, are evident in VOODOO and PEAKO-peakTree based results (Fig 4c, g, h). One notable difference between Fig. 4g and h is the higher liquid reflectivity values for the Ka-band radar MIRA-35 in altitudes above approximately 3 km. This difference of roughly 5 dB can be explained by attenuation due to liquid water, which impacts the 94 GHz system more strongly than the 35 GHz cloud radar, in addition to the higher sensitivity of the

# Punta Arenas, 2019-03-13

**Figure 4.** Radar and lidar observations and retrieved liquid probability and liquid peak reflectivity during the 13 March 2019 case in Punta Arenas. (a) Equivalent radar reflectivity measured by LIMRAD94; (b) Mean Doppler velocity measured by LIMRAD94; (c) probability of cloud droplet presence predicted by VOODOO using LIMRAD94 cloud radar Doppler spectra; (d) attenuated backscatter at 532 nm measured by the Polly$^{XT}$ lidar, overlaid (in pink) with the probability field from (c) where $p_{CD} > 0.65$; (e) number of peaks detected by PEAKO-peakTree in LIMRAD94 spectra. The black boxes A-D mark times/ranges for which example spectra are shown in Fig. 5a-d; (f) number of peaks detected by PEAKO in MIRA-35 spectra; (g) liquid peak reflectivity derived by PEAKO and peakTree from LIMRAD94 spectra, the solid horizontal black lines at 1.2 and 7 km height mark the boundaries of the second LIMRAD94 chirp; (h) liquid peak reflectivity derived by PEAKO and peakTree from MIRA-35 spectra.

(pulsed) 35 GHz cloud radar.

As noted earlier, VOODOO can misclassify Doppler spectra with narrow ice peaks as liquid-containing. In turn, peakTree offers the possibility of analyzing the moments of subpeaks identified as liquid, including the LDR, which can be helpful to rule out that small columnar ice was misclassified as liquid water in the Doppler spectra. For the liquid peaks identified in the MIRA-35 spectra by the PEAKO-peakTree toolkit in Fig. 4f, the LDR consistently ranges below -25 dB (not shown), which is a strong indication that those peaks are indeed being caused by liquid water.

Detecting multiple liquid-containing layers as shown here is a valuable asset for studies on cloud radiative effects (Barrientos-Velasco et al., 2022; Schimmel et al., 2023), as well as for seeder-feeder mechanism studies (Proske et al., 2021). Furthermore, the identification of a liquid peak offers opportunities for retrievals of liquid water content and vertical air motion by using the liquid peak as air motion tracer (e.g. Kalesse et al., 2016). One big strength of the PEAKO-peakTree toolkit for this kind of application is its high flexibility: Both algorithms can be adjusted to the specific cloud radar Doppler spectra at hand, and

spectral polarimetric variables can be included if they are available, but are no prerequisites.

To further illustrate capabilities and limitations of the PEAKO-peakTree toolkit, four examples of liquid-containing spectra of both radar systems are shown in Fig. 5. The times/ranges at which the spectra were selected are marked with black boxes A-D in Fig. 4e. Please note that, for comparability of the spectra measured by LIMRAD94 and MIRA-35, the units of the reflectivity are normalized by the Doppler bin width.

The spectrum in Fig. 5a was observed at around 8:04 UTC at around 410 m range, i.e. in the first LIMRAD94 chirp. Here, the best PEAKO results (i.e. highest similarity score) are obtained when no averaging of neighboring spectra in time and range is performed for LIMRAD94 spectra (Table 1). As a result, the raw and smoothed LIMRAD94 spectra are almost identical. Small differences between -1 and $0\,\mathrm{m\,s^{-1}}$ result from smoothing. For both radars, two peaks are detected, and the peak near $\pm 0\,\mathrm{m\,s^{-1}}$ is classified as "liquid" by peakTree according to the criteria described in section 2.2.2. The spectra shown in Fig. 5b-d are all

within the second LIMRAD94 chirp, where time/ range averaging over 9 spectra results in the highest PEAKO similarity score (see Table 1). This leads to differences in the depicted raw and smoothed spectra, e.g. in Fig. 5b, where the faster-falling peak is significantly smaller and narrower in the averaged and smoothed spectrum than in the raw spectrum. Averaging over neighboring spectra in the temporal and/ or spatial domain can in some cases even lead to peaks not being detected by PEAKO as e.g. in Fig. 5d. Here, the slowest-falling peak in the spectra of both radars is either smoothed below the minimum prominence required for peak detection (MIRA-35) or not visible at all anymore in the averaged and smoothed spectrum (LIMRAD94).

Summarizing, the averaging and smoothing of Doppler spectra is an effective method to reduce the number of unwanted noise peaks. However, this approach also introduces a challenge by possibly rendering genuine signal peaks undetectable. This is a tradeoff between noise reduction and potential signal loss, and intrinsic to signal processing methodologies.

### 4.3 Case study 2: Frozen drizzle and SIP observed during NASCENT

We choose a second case study, from the NASCENT data set, to compare the cloud radar Doppler spectrum peaks detected by peakTree to hydrometeor types observed in situ by HOLIMO. This case serves to illustrate the capabilities and limitations of the proposed PEAKO-peakTree technique for detecting and attributing peaks to different hydrometeor types in a complex sit-

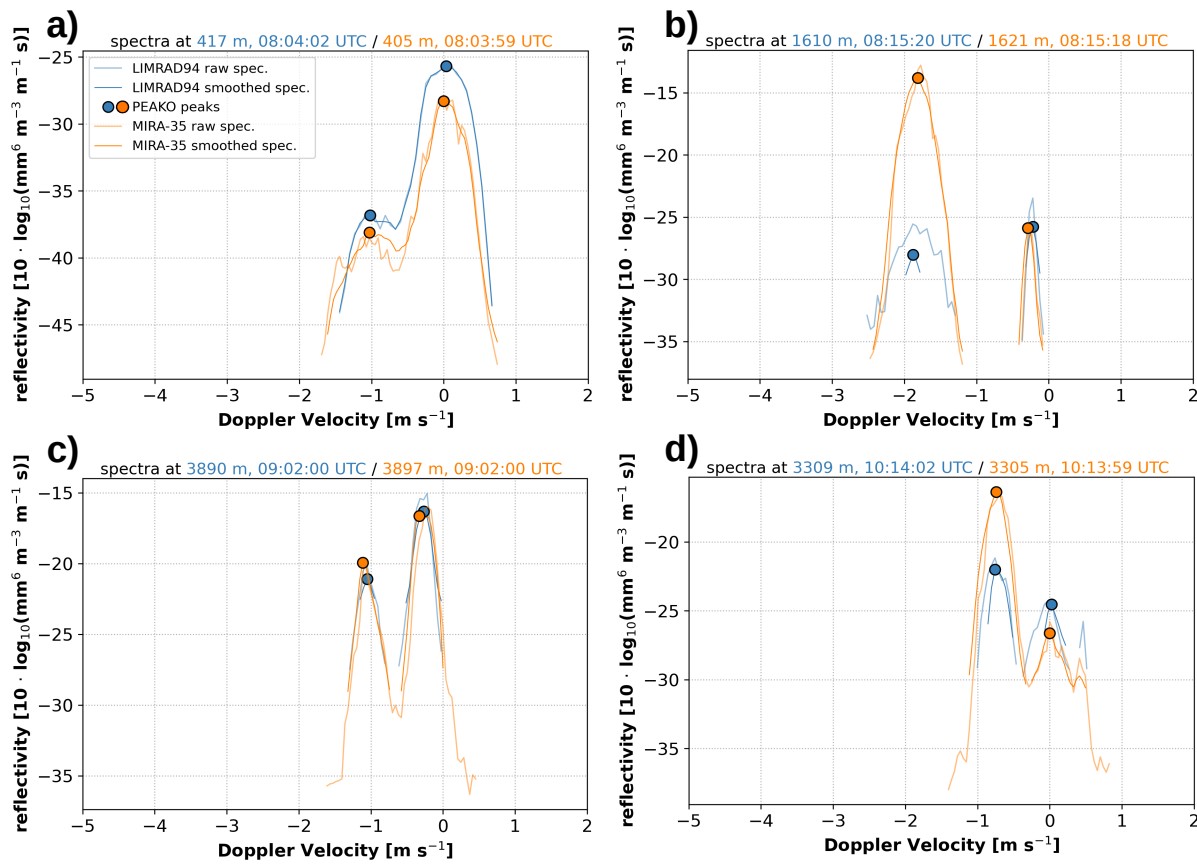

**Figure 5.** Four examples for "liquid-containing" cloud radar Doppler spectra observed by LIMRAD94 (blue lines) and MIRA-35 (orange lines) during the case study presented in Fig. 4. The times/ ranges at which the spectra were picked are marked with rectangular boxes in Fig. 4e. The darker, thinner line represents the averaged-and-smoothed spectrum and the lighter, thicker line the raw spectrum for each radar.

uation. Furthermore, the effect of turbulence and the number of spectral averages $n$, which was outlined in theory in Section 3, is observed and further explored in real radar measurements.

For this purpose, we selected a study period on 12 November 2019 between 15:00 and 16:15 UTC, which has been previously described in detail by Pasquier et al. (2022a). On this day, the weather in Ny-Ålesund was influenced by a warm front and southwesterly winds. A low-level mixed-phase cloud was present, with cloud top height around 2 km and cloud top temperature of approx. -13.5 to -11°C (Fig. 6a). The considered period can be roughly split into two parts: A turbulent period up to around 15:45 UTC (Fig. 6b), presumably favoring the formation of large drizzle drops in updrafts, and a subsequent period

dominated by fallstreaks. The processes responsible for the observed ice particles at the altitude of HoloBalloon, which are suggested by Pasquier et al. (2022a), involve the formation of ice crystals near the cloud top via primary ice production, followed by SIP, which increased their number concentration. The ice crystals then grew, until they were overcoming the updrafts

and falling into the turbulent layer below, where they continued to grow as columns. There, the ice particles were colliding with supercooled drizzle drops, forming "ice lollies" (Keppas et al., 2017) observed by HOLIMO. A sharp increase in the number

of small ice particles ($<100\,\mu$m) was observed during the fallstreak period, which is attributed to SIP via droplet shattering.

Fig. 6 summarizes the JOYRAD94 and HOLIMO observations along the balloon track during the selected case study period, along with the calculated EDR (Fig. 6c) and the number of peaks detected by PEAKO and peakTree (Fig. 6e). Upon initial review of Fig. 6c and e, a link between the number of detected peaks and the EDR becomes apparent. Specifically, a higher number of detected peaks coincides with lower EDR values, while higher EDR values typically correspond to the identification

of one single peak in the Doppler spectra. Based on the findings from section 3, this observed pattern could be explained by turbulence broadening the spectra in the high EDR region, rendering peaks undetectable. However, the number of spectral averages specified in the radar settings could also play a role for successful radar Doppler spectrum peak detection. The number of averaged spectra for the JOYRAD94 chirps are 8, 6 and 4 for the first, second and third chirp, respectively (Table 1), which are all well below the recommended number of approx. 20 (section 3). When examining the radar chirp boundaries in Fig. 6e,

however, no artificial jumps in the number of peaks detected by PEAKO and peakTree become apparent. This means that the different settings between the JOYRAD94 chirps probably do not play a major role for peak detection here.

The HOLIMO in situ observations provide additional insights into the observed pattern in the number of detected Doppler spectrum peaks. Fig. 6f and h show the measured liquid and ice PNSDs, respectively. The HoloBalloon track is drawn in the time-height plots in Fig. 6a-e. Throughout the first part of the case study, i.e. the "turbulent period", HoloBalloon was employed

at a relatively constant altitude of approx. 700 m in a region where EDR was high ($>0.001\,\mathrm{m^2\,s^{-3}}$, Fig. 6c). Shortly after the onset of the "fall streak period" at 15:45 UTC, HoloBalloon was lowered into a less turbulent layer. During the entire remaining time of the selected case, HoloBalloon was flying at altitudes below 400 m.

During almost the entire case study, a bimodal droplet distribution, i.e. a cloud and a drizzle mode, is visible in the liquid PNSD (Fig. 6f). Upon transitioning into the less turbulent layer at around 15:50 UTC, the HOLIMO data reveal an increase in the ice

concentration (Fig. 6h), both due to an increase in small and large ice particles (Fig. 6g and h). The presence of multiple peaks in the cloud radar Doppler spectra in the lower cloud regions (below approx. 400-600 m in Fig. 6e) could thus be explained both by the decrease in EDR, and the increase in the concentration of (large) ice particles.

To explore more deeply why multiple peaks are detected in the spectra at lower ranges after 15:50 UTC, we once again employ PAMTRA forward simulations for three selected times marked with vertical dashed lines in Fig. 6. For these times, we compare

the Doppler spectrum peaks detected by PEAKO and peakTree with the in situ observations and forward-simulated spectra. A similar analysis was done in (Pasquier et al., 2022a, , Fig. 11) for slightly different times of interest. Here, we extend the analysis to the comparison with observed cloud radar Doppler spectra, and the application of PEAKO-peakTree. Fig. 7 shows the observed hydrometeor PNSDs along with observed and forward-simulated spectra for the three comparison times. It should be noted that the vertical air velocity is set to $0\,\mathrm{m\,s^{-1}}$ in the PAMTRA simulations, while the observed spectra can be impacted

by vertical air motions, moving them along the Doppler velocity axis.

The first comparison time (marked with 1 in Fig. 6a) is set at 15:12 UTC, during the "turbulent period", when HoloBalloon was employed at the altitude of the second chirp of JOYRAD94. This time is characterized by high EDR (Fig. 6c), and very low ice

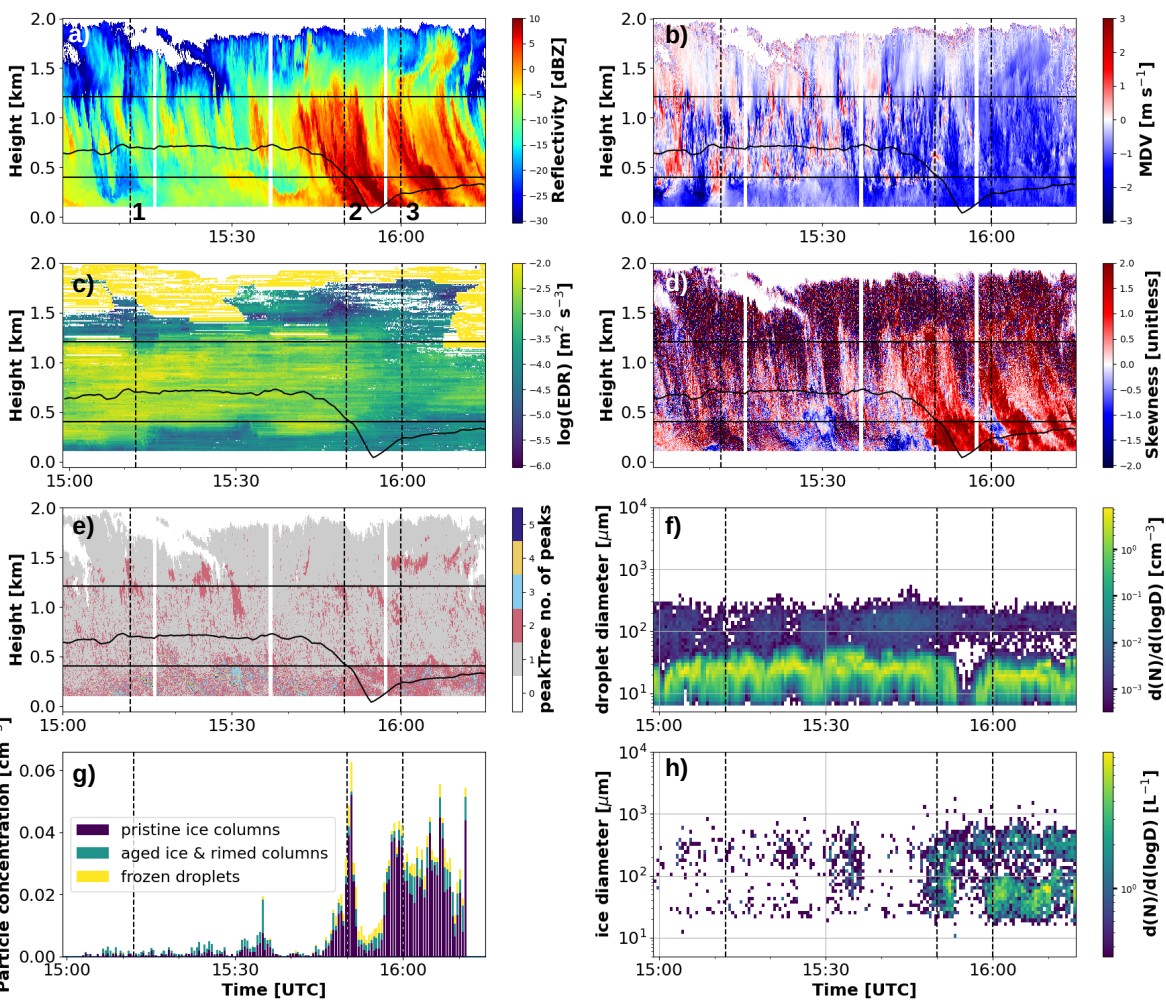

**Figure 6.** Radar and in situ observations during the 12 November 2019 case in Ny-Ålesund. (a) Equivalent radar reflectivity measured by JOYRAD94; (b) Mean Doppler velocity measured by JOYRAD94; (c) Eddy dissipation rate computed from JOYRAD94 observations of Mean Doppler Velocity; (d) Skewness observed by JOYRAD94; (e) Number of peaks detected by peakTree in JOYRAD94 spectra; (f) liquid PNSD measured by HOLIMO; (g) concentrations of main ice habits observed by HOLIMO; (h) ice PNSD measured by HOLIMO (area-equivalent diameter is used); in the radar time-height plots (a-e), the thick solid black line marks the altitude of the HoloBalloon and the horizontal solid lines indicate the boundaries of the chirps of JOYRAD94 as listed in Table 1. The vertical dashed lines mark three times chosen for comparison of the measured cloud radar Doppler spectra, including the peaks detected by PEAKO and peakTree, and in situ observations (Fig. 7).

concentrations (Fig. 6h). The liquid PNSD (Fig. 6f) exhibits a bimodal distribution, corresponding to cloud droplets and drizzle drops. In the JOYRAD94 Doppler spectrum at time 1 (Fig. 7a), only one very broad, slightly skewed peak is detected. The

measured ice and liquid PNSDs for time 1 are shown in Fig. 7d and the resulting forward-simulated radar Doppler spectrum is presented in Fig. 7g. The drizzle mode (purple lines in Fig. 7d and g) dominates the total spectral reflectivity (black line), whereas the individual contribution of the ice PNSD to the total reflectivity spectrum is not discernible. In Fig. 7g, the cloud and drizzle modes are not separable in the forward-simulated spectrum, which, like the exemplary observed radar Doppler spectrum in Fig. 7a, has nonzero skewness and features only one broad peak.

Time 2 is set at 15:50 UTC. During this time, HoloBalloon is lowered through the fallstreaks but still located in the second JOYRAD94 chirp, above 403 m altitude. The observed skewness (Fig. 6d) is positive at time 2, indicating the presence of more than one hydrometeor population. The EDR is still high, and HOLIMO observes an increased concentration of ice particles in addition to the presence of the cloud and drizzle drops (Fig. 7e). The concentration of ice columns and rimed ice is increased compared to time 1, and frozen ice droplets are present. In the resulting forward-simulated spectrum shown in Fig. 7h, again,

supercooled drizzle dominates the total spectral reflectivity, masking the contributions of small ice and cloud droplets. The frozen droplets produce a sub-peak at around -2 to -3 $\mathrm{m\,s^{-1}}$. The example JOYRAD94 spectrum in Fig. 7b features a spike-like peak at -1.9 $\mathrm{m\,s^{-1}}$ and a broad, slightly bimodal peak at slower fall velocities. The small fast-falling peak is lost due to averaging and smoothing, i.e. not visible in the smoothed spectrum anymore and thus not detected by PEAKO-peakTree. Based on the forward simulations, this feature is only attributable to frozen drops due to their strongly negative Doppler velocity. It is not

possible to attribute any of the hydrometeor classes with certainty to the other two modes in the observed spectrum in Fig. 7b, as the fall velocities of ice, drizzle and cloud particles overlap. At this point, we would also like to point out that this particular case illustrates one general limitation of Doppler spectrum-based hydrometeor classification tools: the Doppler spectrum is largely dominated by supercooled drizzle, which constitutes the main peak at approx. -1 $\mathrm{m\,s^{-1}}$. Without the HOLIMO instrument, this peak would have likely been classified as snow particles rather than supercooled drizzle, while the faster-falling emerging mode would have been classified as rimed snowflakes rather than frozen drops.

Finally, time 3 is set at 16:00 UTC, when the tethered balloon is employed in the lowest JOYRAD94 chirp at an altitude of 240 m. This time is characterized by low turbulence and even higher ice concentrations than at time 2, related to heavy SIP occurrence as described in Pasquier et al. (2022a). The skewness (Fig. 6d) is also positive during this time, and 2-3 peaks are detected by PEAKO and peakTree (Fig. 6e), as illustrated by the example JOYRAD94 Doppler spectrum in in Fig. 7c. Two

modes can be separated in the main peak by PEAKO and peakTree, and the liquid cloud droplet peak is visible, but smoothed to a prominence below the detection threshold. The ice peak at large fall velocities corresponds to a fast-falling ice mode, which, in the forward simulations, can be explained by frozen droplets (Fig. 7i). Notably, the ice column mode (orange line in Fig. 7f), which results from the strong SIP, is completely hidden in the drizzle peak in the forward-simulated spectrum in Fig. 7i. At the same time, the liquid peak at the slow edge of the spectrum is detectable in the forward-modeled spectrum.

Summarizing, the presented cases demonstrate clearly how atmospheric turbulence and the superposition of the fall velocities different hydrometeor modes can add complexity to the separability of cloud radar Doppler spectrum peaks. In the first case (time 1, Fig. 7a, d, g), turbulent conditions contribute to the broadening of peaks, rendering the separation of the liquid peak

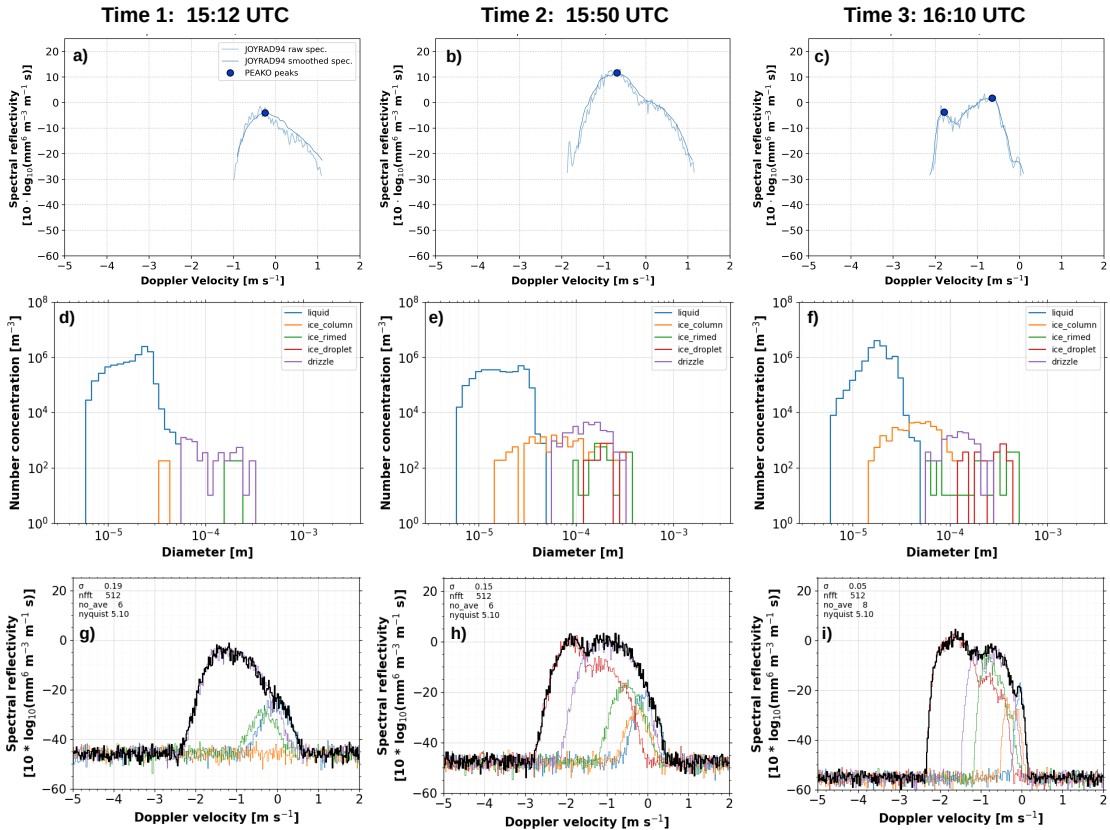

**Figure 7.** Observed and forward-simulated cloud radar Doppler spectra during the 12 November 2019 NASCENT case, along with measured HOLIMO size distributions for five hydrometeor classes. Top row (a–c): Example cloud radar Doppler spectra observed close to the three selected times 1–3 and the HoloBalloon altitude. Center row (d–f): Observed particle number concentrations from the three times 1–3. Bottom row (g–i): PAMTRA forward-simulated spectra using the size distributions from d)–f)

from the drizzle mode impossible. The second case (time 2, Fig. 7b, e, h) showcases how the dominance of the drizzle mode, exacerbated by strong turbulence, can effectively mask other hydrometeor types in the resulting cloud radar Doppler spectra. Conversely, in the third case, lower atmospheric turbulence levels in combination with smaller drizzle and higher ice concentrations facilitate the separability of a fast-falling ice mode and a distinct liquid peak.

Finally, the peakTree selection rules for "fast-falling" and "liquid" populations introduced in section 2.2.2 were applied to the detected peaks of the presented case study. Besides the identification of peaks, peakTree also determines their moments like Ze, MDV, and width, which is helpful for microphysical process studies. To illustrate this, Fig. 8 shows the reflectivities of the selected nodes. Regions of the cloud where EDR is $>0.0001\,\mathrm{m^2\,s^{-3}}$ are represented as gray shaded areas, to highlight where peak detection may be hampered by turbulence. From the HOLIMO observations, we know that liquid droplets were present

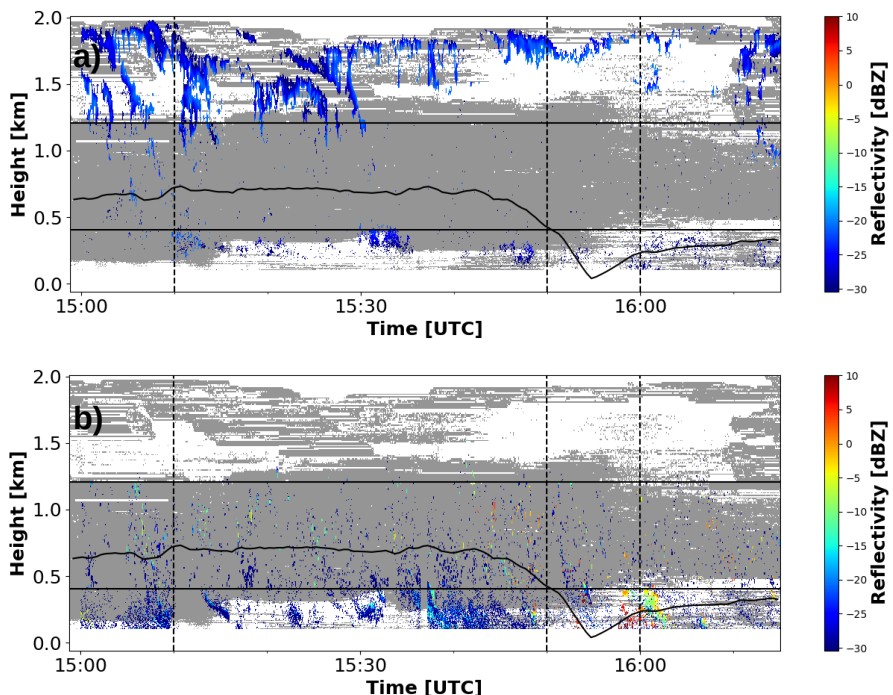

**Figure 8.** Reflectivity of the nodes identified by peakTree as (a) "liquid" and (b) "fast-falling" during the 12 November 2019 case shown in Fig. 6. The grey shading marks high turbulence areas where EDR is above $0.0001\,\mathrm{m^2\,s^{-3}}$. The thick solid black line marks the altitude of the HoloBalloon and the horizontal solid lines indicate the boundaries of the chirps of JOYRAD94 as listed in Table 1. The vertical dashed lines mark the three times chosen for comparison of the measured cloud radar Doppler spectra, including the peaks detected by PEAKO and peakTree, and in situ observations (Fig. 7).

at the HoloBalloon measurement height (Fig. 6f), i.e. it is certain that liquid peaks are missing in these cloud regions. This can be explained by the findings in section 3 and Appendix A: If the number of spectral averages is low ($n < 20$), the liquid peak can only be separated by PEAKO-peakTree for low to moderate turbulence conditions, and if the liquid peak is very prominent in the Doppler spectrum (Fig. A1b).

Nevertheless, some valuable information can be obtained by PEAKO-peakTree in those cloud areas that are not as strongly characterized by turbulence and/ or where the peaks of different hydrometeor populations are well separable: Fig. 8a reveals a liquid cloud top, typical for Arctic mixed-phase clouds, and another liquid-containing layer at around 250 to 300 m height. In previous analyses of this case, it was assumed that supercooled liquid water is present near the cloud top, which is of importance for primary and secondary ice formation processes. With the help of the PEAKO-peakTree toolkit, we are able to confirm the presence of the supercooled liquid cloud top. Furthermore, PEAKO-peakTree expose the sporadic existence a

second liquid-containing layer below cloud top, which is of relevance for ice growth processes like riming, or SIP taking place in the cloud.

The fast-falling node (Fig 8b) is mostly detected in low ranges (below 500 – 600 m). Heavily rimed ice crystals and faceted rime (belonging to the "aged" class, Fig. 6g and Fig. 10 in Pasquier et al. (2022a)) were observed throughout this case, and potentially contribute to the fast-falling Doppler spectrum peaks. Furthermore, based on our forward simulations, we expect that the frozen drops, which were increasingly observed after HoloBalloon was lowered (Fig 6g), contribute strongly to the observed fast-falling peaks. The frozen droplets, including ice lollies, are mainly observed in the time from 15:50 to approximately 16:15 UTC, coinciding with elevated reflectivities (-10 up to approx. 5 dBZ) in the fast-falling node around 16 UTC (Fig. 8b).

When considering the cloud region at around 15:32 between approx. 200–300 m height, the PEAKO-peakTree toolkit successfully detects three distinct peaks in Fig. 6e, which are attributed to the liquid and the fast-falling hydrometeor population (Fig. 8). This example highlights the PEAKO-peakTree toolkit's ability to identify different hydrometeor types and represents a first step towards extracting their properties, which opens doors for detailed cloud microphysical studies. The PEAKO-peakTree toolkit can be applied to data sets without coinciding in situ observations and provide valuable information about the hydrometeors present in the cloud radar observation volume, to some extent even within highly dynamic cloud systems.

## 5   Conclusions

This paper showcases the synergistic application of two algorithms designed for detecting and interpreting peaks in cloud radar Doppler spectra: PEAKO and peakTree. Using them together can be a powerful tool to extract comprehensive information from cloud radar observations. PEAKO is used to determine the optimal parameters for peak detection in cloud radar Doppler spectra, while peakTree is a tool for peak detection, structuring and interpretation. Overall, this study describes recent developments and presents valuable insights into the capabilities and limitations of these two algorithms. Both of them, initially introduced in 2019, underwent substantial progress, including the translation of the PEAKO algorithm from Matlab into Python. Cloud radar Doppler spectra files from RPG-FMCW radars are now supported by both methods, which were originally developed for MIRA-35 and/ or ARM cloud radar data. Moreover, the compatibility of the two algorithms is a notable development, allowing PEAKO peak finding parameters to be passed to peakTree through a configuration file. Furthermore, both algorithms are now openly available on GitHub, facilitating accessibility for the scientific community.

The detection of peaks in cloud radar Doppler spectra is impacted by the radar settings and atmospheric dynamics. To gain a better understanding of the combined effects of turbulence and the number of averaged spectra on the separability of Doppler peaks, simulations with the radar forward operator PAMTRA were performed. The results point to a minimum number of 20 averaged spectra to prevent spurious noise peaks being detected in the spectra. For turbulence broadening exceeding $\sigma = 0.06\,\mathrm{m\,s^{-1}}$, the successful detection of liquid sub-peaks decreased sharply for all simulation runs, including radar settings with up to 120 averaged spectra.

PEAKO $k$-fold cross-validation with up to 2500 spectra as training data, in which the peaks were marked manually by a human

expert labeller, were performed for each of the eight types of radar data used in this study. The results suggest that around 100 to 200 training spectra are sufficient for PEAKO training, as the training and validation curves converge at a constant skill score at approximately this value range.

Two case studies were presented to demonstrate the algorithms' capabilities in detecting and interpreting cloud radar Doppler spectra peaks. In the first case, data from two co-located vertically pointing cloud radar systems was used. PEAKO and peak-Tree identified nearly identical liquid layers in both datasets, which also agreed well with liquid layer structures identifiable in the lidar backscatter and the probability of liquid yielded by the machine-learning based retrieval VOODOO. The second case study offered the opportunity to compare cloud radar Doppler spectra to observations collected by a holographic imager deployed on a tethered balloon flying in the cloud. However, due to intense turbulence and radar settings involving a low number of averaged spectra, peak detection in the observed radar spectra proved challenging. Additional forward simulations using PAMTRA highlighted the strong impact of atmospheric turbulence, broadening the peaks to the extent that sub-peaks became indistinguishable. These findings highlight a critical insight: the performance of the algorithms is notably influenced by the quality of the data, and the influence of atmospheric turbulence is a key factor. Nevertheless, in low-turbulence cloud areas, the algorithms successfully identified multiple peaks, enabling the identification of fast-falling and liquid nodes. This is especially useful for cloud microphysical studies, e.g. of multilayer-clouds in which multiple liquid-containing layers can lead to seeder-feeder effects. Another potential PEAKO-peakTree application is to facilitate and broaden the application of existing Doppler spectrum peak-based retrievals to larger data sets. This includes e.g. retrievals of drop size distributions, liquid and ice water content, or using the liquid peak as air motion tracer (Kalesse et al., 2016).

A logical progression of this work includes the integration of PEAKO and peakTree into Cloudnet, which is currently being developed towards incorporating cloud radar Doppler spectra. Given the high adaptability of both Python-based algorithms, this proposed implementation is expected to be a feasible task with manageable complexity. peakTree's capability to extract reflectivities for each detected node presents an avenue for detailed cloud microphysical process studies. Beyond the identification of fast-falling and liquid nodes as shown in the second case study, other hydrometeor types can be freely defined by setting thresholds for Z, MDV, spectrum width or LDR (for dual-pol radar systems). Implementing PEAKO and peakTree across all Cloudnet sites holds immense potential for research applications aiming at deepening the understanding of cloud microphysical processes.

*Code and data availability.*  JOYRAD94 moment data from the NASCENT campaign is available on Zenodo: Pasquier et al. (2022c)

The JOYRAD-94 Doppler spectra data is available at PANGAEA: Gierens et al. (2023).

PEAKO is available on GitHub at https://github.com/ti-vo/pyPEAKO/

peakTree is available on GitHub at https://github.com/martin-rdz/peakTree

## Appendix A: Peak separability for well-separated bimodal particle populations

The simulations from section 3 were repeated considering only two particle populations to illustrate conditions where the two peaks are more easily distinguishable: We simulated a cloud droplet particle population with a total number of $55.3 \times 10^6 \, \mathrm{m}^{-3}$ and a mean size of $18 \, \mu\mathrm{m}$, resulting in a peak of -23 dBZ at -0.05 ms$^{-1}$ and a slighly rimed ice particle population with $2.9 \times 10^3 \, \mathrm{m}^{-3}$ and a mean size of $294 \, \mu\mathrm{m}$, resulting in a peak of -12 dBZ at -0.84 ms$^{-1}$.

Compared to the simulations based on the HOLIMO observations, the liquid peak possesses a significantly higher reflectivity, while the second peak has a lower reflectivity (-23 dBZ instead of 0 dBZ) and spectral width (0.24 ms$^{-1}$ instead of 0.44 ms$^{-1}$). For peak finding, the PEAKO-trained parameters were used. With increasing $n$, the smoothing span decreases from 0.55 to 0.15 ms$^{-1}$, while the prominence threshold is constant at 0.5 dB. In line with the simulations based on HOLIMO PNSDs, the number of detected peaks with the optimized peak finding parameters is 1 and 2 for high and low turbulence broadening, respectively - regardless of $n$ (Fig. A1 dots). Contrarily, for the fixed peak finding parameters, the number of peaks strongly increases for low $n$ due to (wrong) noise detections (Fig. A1 open diamonds). The liquid droplet peak is reliably detected up to a kinematic broadening of 0.1 ms$^{-1}$, which is slightly higher than for the spectra forward-simulated based on HOLIMO observations. Reliable detections of the liquid peak can even be obtained for $n = 6$ and $n = 10$. At higher $n$, the liquid peak can still be detected for kinematic broadening conditions above 0.125 ms$^{-1}$.

*Author contributions.* TV and HKL developed PEAKO. MR developed peakTree. TV applied PEAKO and peakTree to the cloud radar data. MR performed the PAMTRA simulations. All authors contributed to instrument operation either during the DACAPO-PESO or the NASCENT campaign. All authors contributed to the writing and editing of the manuscript.

*Competing interests.* The authors declare that no competing interests are present.

*Acknowledgements.* We gratefully acknowledge the funding of the German Research Foundation (DFG) in the frame of the special priority program on the Fusion of Radar Polarimetry and Atmospheric Modelling (SPP-2115, PROM, Grant KA 4162/2-1).

LACROS operation was supported by the European Union (EU) Horizon 2020 (ACTRIS; grant no. 654109).

We also gratefully acknowledge the support by the SFB/TR 172 "ArctiC Amplification: Climate Relevant Atmospheric and SurfaCe Processes, and Feedback Mechanisms (AC)[3]" funded by the DFG (Deutsche Forschungsgesellschaft).

We also gratefully acknowledge funding from the Swiss National Science Foundation (SNSF, Grant 200021_175824)

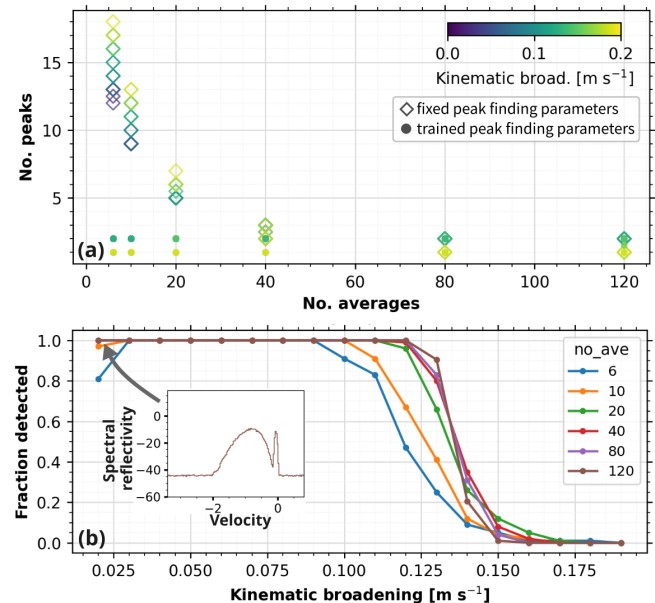

**Figure A1.** PEAKO-peakTree peak detection results in forward-simulated cloud radar Doppler spectra using PAMTRA a) number of detected peaks by PEAKO-peakTree vs. the number of spectral averages $n$ set in the forward simulation. Compared are two sets of peak finding parameters: fixed (indicated by open diamonds; a prominence threshold of $1.5\,\mathrm{dB}$, minimum width of $0.03\,\mathrm{m\,s^{-1}}$, and no smoothing along the velocity dimension) and trained individually for each $n$ (dots). The marker color represents the turbulence broadening in $\mathrm{m\,s^{-1}}$ set in the PAMTRA forward simulation. b) fraction of successfully detected liquid peaks in 100 forward-simulated spectra vs. turbulence broadening in $\mathrm{m\,s^{-1}}$ for radar settings with $n = 6, 10, 40, 80$ and 120. One example of a forward-simulated spectrum is shown as inset ($n = 120$ and $\sigma = 0.02\,\mathrm{m\,s^{-1}}$).

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
