# Peer review of "PEAKO and peakTree: Tools for detecting and interpreting peaks in cloud radar Doppler spectra – capabilities and limitations"

_EGUsphere, 2024_

## Referee Comment (RC1)

**General comments:**

In this manuscript, the authors combine two algorithms (PEAKO and peakTree) for detecting and characterizing peaks in cloud-radar Doppler spectra. Based on two test cases, they investigate the capabilities and limitations of their newly implemented PEAKO-peakTree toolkit. Their results indicate that the presented analysis method can be used to identify supercooled liquid water and ice peaks in mixed-layer clouds as long as turbulence is not too high and radar settings guarantee high-quality Doppler data.

Overall, the manuscript is well written and the results are presented clearly. For easier readability, long compound nouns that are used occasionally (particularly in the abstract) could still be simplified (s. comments below).

I particularly like that the authors make an effort to assess under which conditions the best results can be expected for analyzing multimodal cloud-radar Doppler spectra with the PEAKO-peakTree toolkit. Nonetheless, I would still suggest to include more information about how different factors impact the results and conclusions, as listed in the specific comments below. The description of how turbulence is modeled and included in the analysis should be expanded, in particular, so the readers can better understand this crucial part of the analysis, even if they are not familiar with the cited paper(s) of how turbulence can be modeled and relates to radar Doppler spectra.

Considering the high quality of the study and the novelty of the results, I would therefore suggest to publish the manuscript after these points are addressed.

**Specific comments:**

**Abstract**: Some sentences rely heavily on nominal style with long compound nouns, which makes them difficult to read and understand promptly.

In l. 8, for example, '…use of two cloud radar Doppler spectra peak analysis algorithms' could be replaced by '… use of two algorithms for analyzing the peaks in cloud-radar Doppler spectra' or '… use of two algorithms for analyzing the peaks in Doppler spectra recorded by cloud radar measurements'. Similarly, in l. 10, 'The learned Doppler spectrum peak detection parameters …' could be replaced by 'The optimal parameters derived for detecting all relevant peaks in the Doppler spectra …'.

**Analysis**: What is the impact of clutter (= strong non-meteorological signal, e.g. reflections from the (wet) ground) in the radar data on the analysis results? Do the cloud-radar data have to be free of clutter? Is it not necessary to filter the data before detecting peaks, because the clutter is filtered automatically by the PEAKO peak finding algorithm? Can the analysis also be applied to a different radar with different clutter characteristics?

These points should be addressed briefly somewhere in the manuscript, e.g,. in Section 2.2 and in the Conclusions.

**Section 2.4**: I do not fully understand what is done here. Particularly, how does MDV enter the calculations? From the text, I assume v_h is obtained from the atmospheric models at each radar range bin, but it is not clear how this is then translated into a power spectrum and what the role of MDV is.

Considering the importance of turbulence in the radar analysis later on, this section should be expanded to make it easier to understand how turbulence is estimated for the analysis.

**Section 2.5**: Why do you use a constant fill value for empty size bins and why is a value of 10 m-3 chosen, in particular? Intuitively, I would assume filling the missing values with another value or by spline interpolation could also be a valid approach. Would using different approaches modify the peak characteristics of the forward-modeled Doppler spectra and thus change the results significantly?

**Section 3**: How would the results and conclusions change, if a different prominence threshold and minimum peak width (instead of 1.5 dB, 0.03 ms-1) were chosen? Or, why did the authors select these specific values?

For example, if the peak prominence threshold and/or the peak width threshold for this evaluation were increased, it seems plausible to expect that a lower number of spectral averages will already lead to a reasonable number of identified peaks, i.e. the curves in Fig. 2 reach more realistic values more quickly.

**Section 4.1**: During training, is it important to already include Doppler spectra from the type of precipitation that is to be studied later on, e.g., if we want to identify liquid peaks do we need to include (many) Doppler spectra that have liquid peaks in the training dataset? This would be important to mention, because, then, the training dataset cannot be chosen at random but would require a pre-selection process to obtain a suitable subset of all available data.

Genenerally, how 'realistic' are the peaks identified by a human user and what effect on the results would be observed if the same data were used by different humans to create the training dataset?

**l. 450 ff**: Just by visually comparing Figs. 6c and e, the correlation seems rather vague. Is this correlation evident (more clearly) from plotting log(EDR) vs. the peakTree no. of peaks, and maybe even quantifiable through the Pearson correlation coefficient?

**l. 511 ff**: In Fig. 6h, time 3 is characterized by a higher ice particle concentration (including larger ice particles according to Figs. 7d-f) than times 1 and 2. Could this be the main reason why an additional ice peak can be observed in the Doppler spectrum at this time, and the changes in turbulence only play a minor role here?

**l. 573 f**: This statement seems rather aggressive, considering that no additional information is given on how these applications could actually be achieved from the methods and results presented in this study and no citations are included where radar Doppler spectra are already used for those applications.

**Technical corrections:**

l. 32f: No line break between minus sign and 37 °C. Use minus instead of hyphen.

l. 141: Should the units of the sampling volume be cm^3 instead of cm^-3, i.e. no minus sign?

l. 211: Peak prominence in dB instead of dBZ?

Figure 2: Add the units of velocity and spectral reflectivity to panel c?

l. 359: maybe better rephrase as '… reflectivities of undetected and falsely detected peaks…'

l. 440f: …., with cloud top height around 2 km and cloud top temperature of approx. ….

l. 507 f: Replace compound noun 'hydrometeor mode fall velocities' with 'the fall velocities of different hydrometeor modes' or something similar

l. 551: 'cloud radar Doppler spectrum peak detection …' could be replaced by 'Detecting peaks in cloud-radar Doppler spectra …'

l. 553: maybe use present tense 'point to' instead of past tense 'pointed to' (similar to 'the results suggest that …' in l. 558)

l. 557: … up to 2500 spectra as training data, …

l. 561: co-located instead of closely located

l. 577: 'capability' instead of 'capacity'

---

## Referee Comment (RC3)

This study presents the status of two publicly available tools for the analysis of radar Doppler spectra named PEAKO and peakTree. The tools were first introduced in 2019 but went under appreciable development particularly targeted to allow the synergistic use of them. Despite the work done on the algorithm is rather technical, the authors provide a considerable amount of application examples that greatly elevate the scientific interest of the study.

The manuscript is well-written and easy to follow. The conclusions are derived from the experimental results offered. The data is presented clearly and the quality of the figures is consistent with the highest standards for scientific publication.

Despite the overall quality of the paper I still see a few points that I suggest to address before publishing. The main limitation I see in this paper, considering its intended goal, is the lack of a clear description of the added value of the combined PEAKO-peakTree tool. It is unclear what the two algorithms can do individually, what are the overlapping capabilities, and what kind of benefit (either additional scientific information extracted or reduced effort to use) one can derive from the combination of them. Moreover, I think that the results derived in section 4.3 highlight some limitations of the radar Doppler analysis which I believe are worth a more extended discussion. I believe addressing these points does not require substantial work or additional experiments, but rather some work on the manuscript text. Thus, I recommend the paper to be published after minor revisions.

**1  Major points**

1. Figure 1. I had some difficulties in using this figure to facilitate my understanding of the paper. First, as far as I understood, the training phase of PEAKO requires human intervention. In this case, I would add this to the input block of PEAKO. Further, PEAKO also identifies peaks itself, right? At least in subsequent figures there are PEAKO-found peaks which is not an output of PEAKO in fig.1 I would add the complete list of output of PEAKO to the figure, as well as the output that is not given to peakTree.

2. Section 3. Aren't the results derived from a very limited case study and using specific assumptions regarding particle properties (scattering and fallspeed)? A different case comprising different particles might be easier to manage (i.e. a mixture of ice, snow and graupel/hail) or much more difficult (small ice crystals mixed with supercooled liquid, or snow and drizzle). Also, the assumptions made in term of scattering properties and fallspeed of particles might have a noticeable impact on the peak prominence and separation respectively. Is there a specific reason to use that particular case and simulation setup and derive general conclusions from that? I would suggest exploring some different options in terms of scattering and fallspeed to assess the effect of those assumptions on your conclusions, and perhaps to include a few cases (also as supplementary material) that might be completely fictitious, and not based on measurements, to illustrate some clearly easy separable situations and a very difficult one.

3. This is connected to my previous comment about Fig. 1. It is unclear when PEAKO and peakTree are used as separate instances and when they are combined. Fig. 1 and the discussion about PEAKO seem to suggest that the purpose of this algorithm is to facilitate the job of peakTree, though it is unclear what might have been different for peakTree without the use of PEAKO and it seems that PEAKO is a valuable software by itself for peak-finding and it is also used with this purpose later in the results. The contribution of peakTree to the overall scientific analysis can also be emphasized because sometimes there is a feeling that PEAKO is used to identify peaks and peakTree is left without a clear purpose. This can be addressed in a number of ways:

   - Restructure Fig. 1 as suggested in my previous comment including all input and output of the two software.
   - Clearly write what is the added value of having peakTree ingesting the optimized configuration file of PEAKO. Or alternatively, state what one would have had to do with peakTree without the use of such optimized configuration.
   - Declare at the beginning of subsections 4.2 and 4.3 what is the intended purpose of the use of PEAKO and peakTree. In particular what information is aimed to extract from the capabilities of peakTree to separate various peaks and their moments.

4. Section 4.3 - The analysis of the results seems to describe a situation with a classic Arctic mixed-phase cloud consisting of ice particles with liquid at cloudtop and a more unusual additional liquid layer at the bottom. However, from the HOLIMO observations, the results of the forward simulations and the discussion at lines 480-493 suggest that this cloud, and especially its reflectivity signal, is largely dominated by supercooled drizzle, while the frozen mode is of little importance. Perhaps this is interesting and should be emphasized as a limitation of the spectral analysis tool because I have the feeling that without the HOLIMO instrument, one would have very rapidly attributed the main peak at -1m/s to snow particles instead of supercooled drizzle while the faster falling emerging mode would have been classified as rimed snowflakes.

5. Fig. 7c it is surprising that the algorithm did not catch the supercooled drops peak which is quite obvious in the spectrum. Can you comment a bit on this and declare if you plan work on the algorithm in order to better catch these situations? I also have the feeling that a human would have labeled it and a pure ML algorithm such as PEAKO could have identified it as well. This is particularly discouraging because even the PAMTRA simulation seems to model the Doppler spectrum particularly well. Perhaps, one can envision using PAMTRA as a generator of synthetic Doppler spectra to train PEAKO on a very large number of situations without the need for a human intervention.

**2 Minor points**

1. Line 3 - The phrasing "convoluted by dynamical effects" puzzled me. It suggests that different hydrometeors would produce distinct peaks if not disturbed by turbulence or vertical and horizontal wind shear (combined with radar finite aperture and pulse duration). In reality, various particle spectra would overlap anyway. The caveat of addressing only particles with "sufficient different terminal fall velocities" seems again to not make it clear enough. Particles with clearly distinct peaks can still have significant tails in the velocity distribution and by a change in the total number concentration of one population the second one can appear in the spectrum or be completely masked by the tail of the other population. So, I would try to simplify this statement by clearly stating that Doppler spectra of different hydrometer types overlap and turbulence and other dynamical effects further smear the observed Doppler spectrum. But I understand that I am being very picky.

2. Line 74, 177, 341, 349, 358, 366, 369, 370, 558 - the term "user" is a nice shortcut, but it feels odd. The person that is tasked to label peaks is not really a user. It might be a user of PEAKO, but again, it is not necessary, it could be anyone with sufficient "motivation" to perform such an effort. I would call this capable person: expert, labeller, decisor, identifier, and similar depending on the context. But again, I am very picky

3. Section 3. I do not like the term "preconditions", or, at least, I do not understand these conditions come before what exactly. Why not just call them "conditions"?

4. I think the ice lollipops were first named ice lollies according to Keppas et al. 2017 (https://agupubs.onlinelibrary.wiley.com/doi/full/10.1002/2017GL073441). I know that Pasquier already called them ice lollipops, but it might be better to keep it consistent and avoid possible confusion in the future. Similarly at line 531, the term droplet lollipop seems misleading because, out of context it might be thought to be a strangely-shaped liquid drop.

5. Fig. 6e In the label in the figure says number of peaks detected by peakTree in the caption PEAKO. As long as these are distinct software it is better to clarify which one is responsible for the peak detection.

---

## Referee Comment (RC4)

**Summary:**

This study describes updates to two separate algorithms—PEAKO and peakTree—and demonstrates their synergistic use through two separate case studies. The authors find that the combined algorithms are able to reliably identify multiple Doppler spectra peaks associated with distinct hydrometeor populations (e.g., supercooled liquid water and ice crystals) outside of cases of moderate-to-strong turbulence. The manuscript is generally well written, despite at times there being a lot of observation detail and acronyms to try to keep track of, and the figures are informative as well. I also appreciate the codes being provided to the community on Github. There are a number of things that I think would help improve the manuscript detailed below, but none of them rise to anything major scientifically. I therefore recommend this paper for publication after *minor revisions*.

**Overall comments:**

- The main novelty of this paper seems to be the synergistic use of PEAKO and peakTree, which performs fairly well (at least for the case studies shown). However, on L161 it is described that PEAKO itself has a peak detection function that is used to find the optimal parameter settings compared to human-identified peaks. So in that way, couldn't PEAKO just be used to find and isolate the peaks itself without relying on peakTree? I suspect there are benefits to using peakTree and that it offers more capabilities (e.g., moment information rather than just the peak locations?), but this should be made a bit clearer to the reader up front who may similarly wonder why PEAKO isn't sufficient on its own.

- I admit I might be missing something fundamental here, but it isn't clear to me how the peakTree species identification was decided. On L202, couldn't these parameters also correspond to cloud ice? What is it about these values that denotes only cloud liquid droplets? (This is seemingly agreed with on L394-395). Similarly, on L210, I feel like these criteria would also fit for moderate to relatively large liquid droplets, but here they are attributed only to frozen drops and rimed ice particles.

- More generally, I would be interested in some discussion/acknowledgement of what would happen to the performance of PEAKO-peakTree if non-Rayleigh scattering is present. Has this been investigated? Can it be included in training data? I know this is not the main motivation of the paper, but with Ka- and W-band radars this could reasonably be expected to be somewhat common.

- I would like to see a bit more objectivity regarding the comparison of VOODOO with the attenuated backscatter for the comparison on L389. While there is good agreement in general, there are a number of areas VOODOO picks up on that the lidar does not. Even when there is agreement, the VOODOO probabilities seem to be in much broader layers than the attenuation. Which is more realistic? Is it possible to overlay one field on the other for a more obvious comparison or do something quantitative beyond describing the agreement as "well" (L563)

**Specific comments:**

1. L11: The use of "structure" here as a verb is a bit unclear to me. Can a different word or a more descriptive phrase be used to describe what is actually meant?

2. L173: For identified peaks with very different minima on either side of the peak (say, the leftmost peak out of many), which side is used to define the prominence? The side with the larger minimum value?

3. L179: Are the training data with peaks identified by humans done on data that has been smoothed at all, or is that data raw so that the impact of *any* smoothing/averaging can be evaluated?

4. L202: I might be missing something very fundamental, but it isn't clear to me how these simple criteria couldn't also refer to very small cloud ice particles, which should also have very low Z and terminal velocities. Is there a reason to believe any particles meeting this criteria should only correspond to liquid droplets?

5. L217: If this output is being used to validate PEAKO-peakTree, can a bit more be said about VOODOO's accuracy or when it may fail?

6. L226: I admit that I am not so familiar with this method of estimating EDR, but I would have assumed that EDR is more a function of spectrum width than MDV. At first glance it isn't clear to me why MDV itself would necessarily indicate anything about turbulence. Couldn't enhanced values of |MDV| just indicate updrafts and downdrafts that themselves don't have anything to do with turbulence, or does L232 indicate that it is binned 5-min values of MDV and their variability that are related to turbulence (so more like spectrum width of MDV values rather than the spectra itself)?

7. L251: Where did the value of 10 m-3 come from? How sensitive are the results to this value? I would have assumed interpolation would work better here. Depending on the number concentrations in the surrounding bins, couldn't this value still cause artificial peaks to be detected?

8. L310: Can the authors explain a bit more about why was this done?

9. L368: If there are this many issues with most of the JOYRAD94 chirps such that the training data is unreliable, why should this data even be incorporated into the study?

10. L382: It isn't clear to me how this alone is indicative of multiple embedded liquid water-containing layers. While maybe atypical of these clouds, couldn't this just signify layers of consistent updrafts that offset the MDV there?

11. L402: Given the disagreement between VOODOO/peakTree and the lidar attenuation, is there a reason to believe VOODOO/peakTree over the attenuation? Attenuation in the presence of liquid water seems definitive and I don't know why there wouldn't be any in this region.

12. I find the use of LDR really compelling in this context, as it helps address the point above. I highly suggest the authors include a plot of LDR in the manuscript for readers. More generally, could LDR not be incorporated as part of the training data set (when available)?

13. L451: When I first read "correlation" I assumed in meant positive correlation. It may help readers to clarify "inverse correlation" as it is being introduced.

14. Figure 7: What explains the difference in the Doppler velocity associated with the peak between the observed spectra and the simulated? (E.g., 7a shows a peak at around -0.3 m/s while the simulated peak is near -1.5 m/s due to drizzle).

15. L575: What is Cloudnet? Unless I missed it, this is the first mention of it that I have seen.

16. L580: Would RhoHV not also be useful in this capacity?

**Typos/Grammar/Etc.**

1. L36: "clouds" should be "cloud" at the end of this line.
2. L141: "was" should be "were"
3. L165: It would be clearer if "has meanwhile" was changed to "have now"
4. L271: There is an extra comma after 0.2.
5. L524: "of" should cone after "existence"
6. L580: Should "width" be "spectrum width"?

---

## Author Comment (AC1)

**Responses to Review Comments on "PEAKO and peakTree: Tools for detecting and interpreting peaks in cloud radar Doppler spectra – capabilities and limitations"**

Teresa Vogl     Martin Radenz     Fabiola Ramelli     Rosa Gierens

Heike Kalesse-Los

August 16, 2024

We thank all four reviewers for their time and appreciate the effort they made in carefully reading and evaluating our manuscript. We think that their constructive comments helped us to improve our manuscript significantly. In the following, we want to address the issues they raised in a point-by-point way.

**Reviewer 1**

**Comment 1.1:** *In this manuscript, the authors combine two algorithms (PEAKO and peakTree) for detecting and characterizing peaks in cloud-radar Doppler spectra. Based on two test cases, they investigate the capabilities and limitations of their newly implemented PEAKO-peakTree toolkit. Their results indicate that the presented analysis method can be used to identify supercooled liquid water and ice peaks in mixed-layer clouds as long as turbulence is not too high and radar settings guarantee high-quality Doppler data.*
*Overall, the manuscript is well written and the results are presented clearly. For easier readability, long compound nouns that are used occasionally (particularly in the abstract) could still be simplified (s. comments below).*
*I particularly like that the authors make an effort to assess under which conditions the best results can be expected for analyzing multimodal cloud-radar Doppler spectra with the PEAKO-peakTree toolkit. Nonetheless, I would still suggest to include more information about how different factors impact the results and conclusions, as listed in the specific comments below. The description of how turbulence is modeled and included in the analysis should be expanded, in particular, so the readers can better understand this crucial part of the analysis, even if they are not familiar with the cited paper(s) of how turbulence can be modeled and relates to radar Doppler spectra. Considering the high quality of the study and the novelty of the results, I would therefore suggest to publish the manuscript after these points are addressed.*

**Response:** We thank the reviewer for carefully reading our work and for giving constructive feedback, which we gladly incorporated into the updated version of our manuscript. We are addressing their specific comments in a point-by-point way below. We are also especially grateful for the list of technical corrections, which we gladly incorporated in the updated version of our manuscript.

**Comment 1.2:** *Abstract: Some sentences rely heavily on nominal style with long compound nouns, which makes them difficult to read and understand promptly. In l. 8, for example, '...use of two cloud radar Doppler spectra peak analysis algorithms' could be replaced by '... use of two algorithms for analyzing the peaks in cloud-radar Doppler spectra' or '...use of two algorithms for analyzing the peaks in Doppler spectra recorded by cloud radar measurements'. Similarly, in l. 10, 'The learned Doppler spectrum peak detection parameters ...' could be replaced by 'The optimal parameters derived for detecting all relevant peaks in the Doppler spectra ...'.*

**Response:** Thanks for pointing us to this deficiency in the language/ style of our text. We went through the entire manuscript and tried to replace long compound nouns with other expressions. We think that this helped to improve the readability of the paper and are very grateful for this hint.

**Comment 1.3:** *Analysis: What is the impact of clutter (= strong non-meteorological signal, e.g. reflections from the (wet) ground) in the radar data on the analysis results? Do the cloud-radar data have to be free of clutter? Is it not necessary to filter the data before detecting peaks, because the clutter is filtered automatically by the PEAKO peak finding algorithm? Can the analysis also be applied to a different radar with different clutter characteristics? These points should be addressed briefly somewhere in the manuscript, e.g,. in Section 2.2 and in the Conclusions.*

**Response:** We thank the reviewer for pointing us to the topic of clutter, which is indeed relevant for vertically-pointing cloud radar Doppler spectra in several respects: Ground clutter is usually due to stationary ground structures such as buildings, trees and power lines and results in Doppler spectrum peaks centered at $0\,\mathrm{m\,s^{-1}}$ (Williams et al., 2018). These peaks, which are due to non-meteorological targets, can in principle be filtered easily by peakTree, e.g. by defining a peak selection rule for peaks with low reflectivity and spectral width, and centered at around $0\,\mathrm{m\,s^{-1}}$ Doppler velocity. Only if

[Figure]

Figure 1: Fig. 3c from Williams et al. (2018) showing a typical Doppler spectrum with ground clutter peak at $0\,\mathrm{m\,s^{-1}}$, and their proposed interpolation method (red dots) to retrieve the clutter-free spectrum

the meteorological signal approaches $0\,\mathrm{m\,s^{-1}}$ as well, a problem can arise for extracting the moments of the meteorological signal (see Fig. 1), because peakTree splits the Doppler spectrum peaks at the minimum. This means that integration is done only from the minimum value to the outer edge of the main (meteorological signal) peak, which would e.g. result in a too low reflectivity value. For such cases, an interpolation method such as the one proposed by Williams et al. (2018), sketched by the red dots in Fig. 1 is better suited to remove the clutter signal. Another type of clutter which is often present in vertically-pointing cloud radar data is due to insects (e.g. Luke et al., 2008). Fortunately, the Cloudnet classification mask (Illingworth et al., 2007; Tukiainen et al., 2020) provides a convenient means of distinguishing between cloud, precipitation and non-meteorological targets and can be used as a pre-processing step to only select meteorological targets in the peakTree analysis. As a side note, current work in our group actually focuses on applying peakTree to Doppler spectra marked as "insects" by the Cloudnet algorithm to derive insect number concentrations in the atmosphere. In the data sets presented in this paper, however, neither ground nor insect clutter played a role. In the case of the MIRA-35 cloud radar, a clutter fence around the antenna efficiently suppresses the ground-clutter signal. Insects did not impact the observations due to the locations of the two selected sites (i.e. the Arctic and Patagonia close to the sea with strong winds and low temperatures). However, we agree with the reviewer that mentioning the implications of clutter in the paper is reasonable, and we included the following in the revised manuscript version at the end of Section 2.2:
"In this respect, peakTree can in principle also be used to filter ground clutter, which usually emerges as Doppler spectrum peaks centered at $0\,\mathrm{m\,s^{-1}}$ (Williams et al., 2018). However, in both data sets used in this study, clutter did not play a role, i.e. a removal of peaks due to non-meteorological targets was not required."

**Comment 1.4:** *Section 2.4: I do not fully understand what is done here. Particularly, how does MDV enter the calculations? From the text, I assume $v_h$ is obtained from the atmospheric models at each radar range bin, but it is not clear how this is then translated into a power spectrum and what the role of MDV is. Considering the importance of turbulence in the radar analysis later on, this section should be expanded to make it easier to understand how turbulence is estimated for the analysis.*

**Response:** We thank the reviewer for pointing us to this shortcoming of the manuscript. We should elaborate more on how the time series of MDV measured at a constant range above the radar is being converted to an energy spectrum. This involves a discrete Fourier Transform, as well as transition from time to spatial frequencies by application of Taylor's hypothesis (Taylor, 1938):

$\nu = f/v_h$

This is also the point where $v_h$ enters the picture. We are afraid that our explanation was not very well understandable and propose to restructure the section explaining the EDR retrieval as follows:

"We are using the eddy dissipation rate (EDR, $\epsilon$) as a measure of atmospheric turbulence to identify regions in the cloud where turbulence potentially affects the detectability of sub-peaks in cloud radar Doppler spectra. The EDR is the rate at which turbulent kinetic energy cascades in the atmosphere from large to smaller and smaller eddies, until it is converted from mechanical into thermal energy at the molecular level (Foken, 2008). It is a universal measure of turbulence, i.e. a large EDR means that the atmosphere is dissipating energy quickly, and that the atmospheric turbulence level is high. EDR can be retrieved from cloud radar measurements of MDV as follows (Borque et al., 2016; Griesche et al., 2020): The time series of MDV measured at one radar range gate in a 5-minute time interval is converted to a power spectrum using a Fast Fourier Transform (FFT). Furthermore, a transition from time to spatial frequencies is done assuming Taylor's hypothesis of frozen (isotropic) turbulence (Taylor, 1938):

$$\nu = f/v_h \tag{1}$$

where f is the frequency and $\nu$ is the wavenumber $(m^{-1})$, which can be related to a characteristic turbulent structure size, or eddy length scale $\lambda$, via

$$\lambda = 2\pi/\nu \tag{2}$$

$v_h$ is the horizontal advection wind speed. To estimate $v_h$, wind data from the European Centre for Medium-Range Weather Forecasts (ECMWF) Integrated Forecast System (IFS) are used for the DACAPO-PESO data and wind data from the icosahedral non-hydrostatic (ICON) model for the NASCENT data set. For retrieving the EDR from the resulting spectrum, the following assumption is made: In the inertial subrange, turbulence is assumed to be isotropic and the decrease in energy is proportional to $f^{-5/3}$. The power spectrum S of turbulent energy dissipation can thus be represented as:

$$S(\nu) = A\epsilon^{2/3}\nu^{-5/3} \tag{3}$$

With $A = 0.5$ the Kolmogorov constant. Consequently, if the spectrum derived from measured MDV can be expressed in this way, the EDR $\epsilon$ can be inferred. Following Griesche et al. (2020), linear least-square regressions of the spectrum in 34 different wavenumber $\nu$ intervals are performed, yielding intercept $i$ and slope $s$ parameters. If $s$ is within $-5/3 \pm 1/3$, the fit is considered "good" and EDR is calculated according to:

$$\epsilon = \left(\frac{10^i}{A}\right)^{3/2} \tag{4}$$

If the $s = -5/3 \pm 1/3$ criterion is met by more than one of the 34 intervals, $\epsilon$ is calculated using $i$ from all of these intervals and the mean $\epsilon$ is used."

**Comment 1.5:** *Section 2.5: Why do you use a constant fill value for empty size bins and why is a value of $10\,m^{-3}$ chosen, in particular? Intuitively, I would assume filling the missing values with another value or by spline interpolation could also be a valid approach. Would using different approaches modify the peak characteristics of the forward-modeled Doppler spectra and thus change the results significantly?*

We agree that the treatment of missing values in the PNSD should be up for discussion. The gap filling is somehow a philosophical question, i.e. the question whether the multi-modality observed in situ is a real feature of the PNSD, or if it is caused by sampling only an air volume of $0.005519$ m$^{-3}$ during the 60 s interval (as opposed to the radar observation volume of 2000m$^3$ at 500 m range).

We tested different fill values and methods (including interpolation), but found that this rather simple and low threshold was sufficient. Figure 2 shows the application of three different fill values

[Figure]

Figure 2: PNSDs (lower row panels) with different fill values and the corresponding forward-simulated spectra (upper row panels): (a) forward-simulated spectrum resulting from the PNSD in (d), without fill value ($n_{min}=0\,\mathrm{m}^{-3}$); (b) forward-simulated spectrum resulting from the PNSD in (e), with fill value $n_{min}=10\,\mathrm{m}^{-3}$; (c) forward-simulated spectrum resulting from the PNSD in (f), with fill value $n_{min}=100\,\mathrm{m}^{-3}$

and illustrates the effect it has on the forward simulated spectrum. To answer the question of the reviewer in brief: Yes, the choice of the fill value strongly influences the peak characteristics.
However, we argue that this does not matter fundamentally: The goal here is not a closure with the real spectra measured by JOYRAD94, but a relative comparison of forward-simulated spectra in order to gain an understanding of how the number of averages $n$ and turbulence might affect the peak detection capability of our toolkit. Since all spectra are affected by the filling in the same way, this should overall not impact the results and the conclusions we draw from the simulation study.
Note, that is fill value is only applied between the lowest and highest size bin with actual counts in the respective particle population.

**Comment 1.6:** *Section 3: How would the results and conclusions change, if a different prominence threshold and minimum peak width (instead of 1.5 dB, 0.03 m s$^{-1}$) were chosen? Or, why did the authors select these specific values? For example, if the peak prominence threshold and/or the peak width threshold for this evaluation were increased, it seems plausible to expect that a lower number of spectral averages will already lead to a reasonable number of identified peaks, i.e. the curves in Fig. 2 reach more realistic values more quickly.*

**Response:** This is indeed a very good and justified question. Thanks for raising this important point. We agree that it would have been more "correct" to train PEAKO on the PAMTRA-modeled spectra instead of fixing the prominence threshold, minimum peak width and span for smoothing for all values of $n$.
So, in order to answer this question, we trained PEAKO on forward-simulated Doppler spectra for each $n$. Spectra were generated using PNSDs with different particle habits (as measured by HOLIMO) as a basis: To produce spectra for training with different numbers of peaks, single particle habits were removed from the input for the simulation, resulting in spectra having up to 7 peaks.
Fig. 3 shows the training result for the span parameter for the five PEAKO configurations yielding the highest similarity score per considered $n$. For these simulations, averaging over time and range was by design not possible, so the only PEAKO parameter that can smooth the spectra (and get rid off noise peaks) is the smoothing span. As the number of spectral averages $n$ increases, the optimum span for smoothing decreases, because less spurious noise peaks that require smoothing occur in the spectra.

[Figure]

Figure 3: box-and-whisker plots of the span parameter yielded by the five best (i.e. with the highest similarity scores) PEAKO parameters. Training was performed for each number of spectral averages $n$, which were forward simulated by PAMTRA.

Fig. 4 shows how the updated version of Fig. 2, where it becomes apparent what changes when optimized peak finding parameters are used instead of the fixed parameters. With the optimized parameters, the number of detected peaks does not increase anymore as the number of averages decreases (Fig. 4a), which makes sense, as the increased smoothing span smoothes out the noise peaks.

Furthermore, we added a similar figure to the appendix of the paper, where the same simulations were performed for a more simple Doppler spectrum featuring a more prominent liquid peak (see answer to comment 3.3).
Section 3 has been restructured and rephrased in wide parts, incorporating these new findings:

[revised manuscript text omitted]

**Comment 1.7:** *Section 4.1: During training, is it important to already include Doppler spectra from the type of precipitation that is to be studied later on, e.g., if we want to identify liquid peaks do we need to include (many) Doppler spectra that have liquid peaks in the training dataset? This would be important to mention, because, then, the training dataset cannot be chosen at random but would require a pre-selection process to obtain a suitable subset of all available data. Genenerally, how 'realistic' are the peaks identified by a human user and what effect on the results would be observed if the same data were used by different humans to create the training dataset?*

**Response:** We thank the reviewer for raising this point. It is indeed important to select the training data according to the targets to be studied in the envisioned PEAKO-peakTree application. We suggest to include the following explanatory sentences in Section 2.1.2, to address this point:

“When selecting training data, care should be taken to ensure that the type of (meteorological) targets to be studied later with peakTree are adequately represented in the data set. For example, if liquid water peaks are the focus of the peakTree study, it should be ensured that liquid-containing spectra are included in the training data set. Furthermore, it may be practical to use classification algorithms such as the Cloudnet target classification (Illingworth et al., 2007; Tukiainen et al., 2020), to include only data classified as ”cloud”, ”rain” or ”insects” in the training data set, depending on the envisioned application.”

With regard to the second part of the question, i.e. how realistic are the human-identified peaks: It is difficult to answer this question definitively. In the course of developing PEAKO, several ”human users” or ”identifiers” (see comment 3.8 for the discussion about this term) have marked peaks in cloud radar Doppler spectra and in our experience, the peaks identified by different humans (who are experienced in working with cloud radar Doppler spectra) do not differ much.

**Comment 1.8:** *l. 450 ff: Just by visually comparing Figs. 6c and e, the correlation seems rather vague. Is this correlation evident (more clearly) from plotting log(EDR) vs. the peakTree no. of peaks, and maybe even quantifiable through the Pearson correlation coefficient?*

**Response:** We thank the reviewer for pointing us to this inaccuracy in the text. ”correlation” was a rather unfortunate choice of words. We propose to replace the word ”correlation” with ”link”:
”Upon initial review of Fig 6c and e, a link between the number of detected peaks and the EDR becomes apparent.”

**Comment 1.9:** *l. 511 ff: In Fig. 6h, time 3 is characterized by a higher ice particle concentration (including larger ice particles according to Figs. 7d-f) than times 1 and 2. Could this be the main reason why an additional ice peak can be observed in the Doppler spectrum at this time, and the changes in turbulence only play a minor role here?*

**Response:** We thank the referee for raising this question, whether the occurrence of multiple peaks at time 3 can be mostly explained by the higher EDR during time 3 compared to time 2, or whether the additional ice could actually play a more important role. Indeed, we have previously discussed this point, and have reached the following conclusion: When considering the radar reflectivity and skewness (Fig. 6a and d), it becomes apparent that larger ice crystals should already be present above the low EDR layer (Fig. 6c), i.e. above 400 m e.g. in the fall streaks. In these regions, however, hardly any spectra with more than one peak are detected (Fig. 6e). This means that lower EDR is likely responsible for the detection of more than one peak.

To investigate this issue further, we have performed some additional simulations: The turbulent

[Figure]

Figure 5: Forward simulated Doppler spectra based on the PSDs measured at times 2 and times 3. (a) spectrum based on PSD at time 2 with the observed turbulent broadening ($\sigma$=0.15 m s$^{-1}$; (b) spectrum based on PSD at time 3 with the observed turbulent broadening ($\sigma$=0.05 m s$^{-1}$); (c) spectrum based on PSD at time 2 with the turbulent broadening observed at time 3; (d) spectrum based on PSD at time 3 with the turbulent broadening observed at time 2

broadening for time 2 was decreased to 0.05 m s$^{-1}$, i.e. the value observed during time 3 (Fig. 5c). Similarly, the turbulent broadening during time 3 was increased to 0.15 m s$^{-1}$ (Fig. 5d), as observed during time 2. When decreasing the turbulence level at time 2, the faster-falling ice peak emerges in the simulation (Fig. 5c). However, the large ice peak from time 3 is indeed still visible in the high-turbulence simulation (Fig. 5d). This points at a combination of turbulence and a more prominent large ice mode being responsible for the increased number of peaks in Fig. 6e.

We modified l 511ff as follows:

"Conversely, in the third case, lower atmospheric turbulence levels in combination with smaller drizzle and higher ice concentrations facilitate the separability of a fast-falling ice mode and a distinct liquid peak." Earlier in the manuscript, we state:

"The presence of multiple peaks in the cloud radar Doppler spectra in the lower cloud regions (below approx. 400-600 m in Fig. 6e) could thus be explained both by the decrease in EDR, and the increase in the concentration of (large) ice particles."

**Comment 1.10:** *l. 573 f: This statement seems rather aggressive, considering that no additional information is given on how these applications could actually be achieved from the methods and results presented in this study and no citations are included where radar Doppler spectra are already used for those applications.*

**Response:** We thank the referee for sharing their concerns with our statement in the conclusions, "This is especially useful for cloud microphysical studies, e.g. of multilayer-clouds in which multiple liquid-containing layers can lead to seeder-feeder effects. Other potential applications would be retrievals of drop size distributions, liquid and ice water content, or using the liquid peak as air motion tracer."

In this case, however, we disagree with the reviewer. In our opinion, the conclusions section is also intended to place the presented work in the context of current research, and provide an outlook on its potential future use. The listed applications are indeed quite realistic, and similar studies have been performed where PEAKO-peakTree could broaden or facilitate the application of suggested retrievals. E.g. the liquid peak has been used as an air motion tracer and for retrieving the liquid DSD and LWC by Kalesse et al. (2016) in a case study of Arctic mixed-phase clouds.

We suggest to alter the sentence as follows:

"Another potential PEAKO-peakTree application is to facilitate and broaden the application of existing Doppler spectrum peak-based retrievals to larger data sets. This includes e.g. retrievals of drop size distributions, liquid and ice water content, or using the liquid peak as air motion tracer (Kalesse et al., 2016)."

**Reviewer 2**

**Comment 2.1:** *Doppler spectra observations are particularly useful for interpretating hydrometeor populations and tracing their changes in the air. This work aims to combine previously developed tools, PEAKO and peakTree, to characterize hydrometeor populations and to demonstrate their applicability. However, I am not convinced that there is a real need for combining these two algorithms. This manuscript is way too tedious, and I do not see much new contributions to the community. Please see my comments below.*

**Response:** We thank the referee for openly sharing their concerns. To avoid that future readers might get similar impressions, we decided to emphasize the advantages of the combination of both algorithms more clearly in the manuscript. We do appreciate their detailed comments below and tried to address the issues raised. However, we do not see how the referee comes to the harsh conclusion.

**Comment 2.2:** *The core of spectral analysis is the peak identification which facilitates separations of different hydrometeor populations. Actually, the essential contributions of PEAKO and peakTree algorithms are overlapped. PEAKO uses ML to identify spectral peaks, while peakTree uses a binary tree structure. The authors claimed that the two algorithms are combined. However, one may simply use PEAKO to do hydrometeor population separation without using peakTree. Therefore, I do not see any additional benefits of using peakTree. I do not really see the novelty or the value of publishment of this point. To me, this work is simply the evaluation of PEAKO. However, this group has already developed and validated some ML methods for spectral peak identification in many papers, do you really need an additional paper demonstrating the validation?*

**Response:** We thank the reviewer for sharing their concerns. Similar points were indeed raised by two other referees, and addressed in the responses to their comments (see comment 3.4). In brief, we have spent some work to make the individual areas of application of each algorithm clearer in the updated version of the manuscript.

- The flowchart has been updated to make this clearer

- The algorithm descriptions were updated to emphasize the advantages of using PEAKO and peakTree in combination

While both algorithms can be used standalone, there is a clear advantage in using them in combination: peakTree is much better optimized in terms of computing time, while PEAKO offers a means to optimize the peak detection function in a less subjective "try-and-error" way than it was previously done in peakTree standalone applications.

Regarding the comment that "this group has already developed and validated some ML methods for spectral peak identification in many papers", we disagree. After PEAKO and peakTree were published in 2019, a lot of development work has gone into them, which has not been documented in the

meantime. While the next step clearly is the application of the combined algorithm toolkit to longer data sets, we here validate their synergy, and their application to other radar systems than those for which they were developed.

**Comment 2.3:** *In spite of my criticism on the novelty, I do see the need of assessing the impact of spectral average on peak identification. Unfortunately, this part is very poorly addressed. I list some technical questions as below. If they are well addressed upon major revisions, I would like to see its publication. The number of averaged spectra in Table 1 is not clearly defined, how is it different from the number of coherent integrations?*

**Response:** coherent averaging is averaging before the FFT. Incoherent averaging is averaging of spectra after the FFT. The number of averaged spectra (which was denoted with $i$ in the previous version of our manuscript) is the number of **incoherent averages** performed after the FFT by the internal radar software.
We agree that this is not well defined in our manuscript and went through the text to revise this issue. Indeed, we spotted one wrong occurrence of "coherent" instead of "incoherent", which we corrected. In this process, we also noticed that $i$ was also used for the intercept in the EDR retrieval, so we decided to instead represent the number of averaged spectra with $n$.
In Section 3, we rephrased the part explaining the impact of too low $n$ as follows:
"Moreover, instrument settings not optimized for peak detection in Doppler spectra might result in spectra that are too noisy to detect the subpeaks. If the number of incoherent averages $n$, i.e. the number of spectra averaged after the FFT, is set too low, this results in strong phase noise signatures, which are a challenge for multi-peak analysis. (Zrnić, 1975)."

**Comment 2.4:** *The PAMTRA simulation results in Section 3 shows that this parameter will affect the of spectral peak detection, and smoothing the spectrum in a desirable interval could help to suppress noise peaks. However, the recommended average number is based on the simulation for fixed PAMTRA parameters, in other words, the particle size distribution and atmospheric dynamics do not change.*

**Response:** True, the PAMTRA simulations are based on a "static" PNSD, assuming temporal homogeneity in the averaging period, which is however very short (cf. integration times in Table 1). We believe this assumption to be reasonable.
Comment 1.6 and our response might be also of interest to the reviewer, as it is related to this issue: There, we train PEAKO for forward-simulated PAMTRA spectra with varying $n$ and look into the effect of $n$ on the optimal smoothing span.

**Comment 2.5:** *Do the simulation only represent the effect of Gaussian noise in the simulated Doppler spectra?*

**Response:** We have to admit that we do not fully understand what the reviewer's question is aiming at here. We provide an explanation of the treatment of noise in PAMTRA, hoping that it answers their question:
PAMTRA considers two kinds of noise: The first is the system noise, or clear sky noise, which is instrument specific and has an $R^2$ dependency (due to the increasing observation volume with increasing distance from the radar), R being the range over the radar. This is set in PAMTRA via the input value "pnoise_0", which is defined as the noise level at R=1 km above the radar.
The second type of noise is the speckle (or phase or random) noise, and in PAMTRA, it is implemented via a random number generator following Zrnić (1975). The PAMTRA code is publicly available on GitHub, and we would like to refer to the part where random noise is added to the spectra here: https://github.com/igmk/pamtra/blob/f23ef6899da8f181b507fe422831eedcd9b8b189/src/radar_simulator.f90#L373

**Comment 2.6:** *These parameters can vary greatly over time in actual observations, in that case, additional averaging of Doppler spectra may instead lose useful signals. How should we measure whether to average the spectrum under different cloud environments? Also, the spectral resolution will affect the effect of averaging. How different spectral resolutions will affect the results?*

**Response:** We do not understand what the reviewer means with "these parameters". The Gaussian noise and the PNSD?

Different spectral resolutions, like the other radar settings that can differ between radar instruments (and chirps), are covered by the PEAKO-peakTree methodology. If this parameter is changed, a new training phase is required to obtain the optimum peak finding parameters.

The effect of different cloud environments on the optimum peak finding parameters has not been evaluated yet. This is a point that was also raised by another referee (comment 1.7), and we would also like to point the reviewer to the corresponding answer there.

**Comment 2.7:** *Spectral averaging is entangled with the effect of turbulence. How do you separate their roles in changing the spectral shape respectively in real observations?*

**Response:** True, spectral averaging and turbulence are indeed entangled. However, the scope of this paper (and the therein presented methodology) is not to definitely separate the roles of the two. The combined effects of turbulence and the number of spectral averages are however discussed in great length in Section 3, and we would like to point to comments 1.6 and 3.3 the corresponding answers, where updates to this section are explained.

**Comment 2.8:** *Fig.7(g-i) gives the PAMTRA simulated spectra based on the size distribution measured by HoloBallon, dose the simulations take into account the effect of horizontal wind on the spectra? In the simulations of bimodal distribution, the relative spectral reflectivity of ice peak and liquid peak is different from that observed by cloud radar. The responsible factors should be clearly discussed.*

**Response:** The simulations do take into account the broadening of the Doppler spectrum caused by the horizontal wind. Shifts of the spectrum caused by vertical air motion are not considered, which is stated in the paper. Again, we would like to emphasize that the aim of our paper is not a closure of simulated and observed spectra.

**Comment 2.9:** *The misinterpretation of ice columns to SIP. You may see the presence of ice columns, however, they are not necessarily produced by SIP. If you wish to identify a SIP event, you need to compare Ncolumn with NINP, as did in (Li et al., ACP, 2021; Wieder et al., ACP, 2022; Billault-Roux et al., ACP, 2023).*

**Response:** We agree that this is generally true, the occurrence of ice columns does not necessarily mean that SIP happens, however, this case has been previously analyzed and identified as an SIP event by Pasquier et al. (2022). In this work, the number of INP measured near the ground at three different sites was compared to the number of pristine ice crystals observed by the HOLIMO instrument and it was concluded that SIP must have played an important role to explain the number of small pristine ice crystals, which were observed in the cloud. We refer to this paper in the description of the case in the beginning of Section 4.3, and briefly summarize their findings:

"The processes responsible for the observed ice particles at the altitude of HoloBalloon, which are suggested by Pasquier et al. (2022), involve the formation of ice crystals near the cloud top via primary ice production, followed by SIP, which increased their number concentration. The ice crystals then grew, until they were overcoming the updrafts and falling into the turbulent layer below, where they continued to grow as columns. There, the ice particles were colliding with supercooled drizzle drops, forming "ice lollies" (Keppas et al., 2017) observed by HOLIMO. A sharp increase in the number of small ice particles ($<100\,\mu$m) was observed during the fallstreak period, which is attributed to SIP via droplet shattering."

In our opinion, no further explanation of how SIP was identified is needed, as the focus of our paper is quite different.

**Reviewer 3 (Davide Ori)**

**Comment 3.1:** *This study presents the status of two publicly available tools for the analysis of radar Doppler spectra named PEAKO and peakTree. The tools were first introduced in 2019 but went under*

*appreciable development particularly targeted to allow the synergistic use of them. Despite the work done on the algorithm is rather technical, the authors provide a considerable amount of application examples that greatly elevate the scientific interest of the study.*

*The manuscript is well-written and easy to follow. The conclusions are derived from the experimental results offered. The data is presented clearly and the quality of the figures is consistent with the highest standards for scientific publication.*

*Despite the overall quality of the paper I still see a few points that I suggest to address before publishing. The main limitation I see in this paper, considering its intended goal, is the lack of a clear description of the added value of the combined PEAKO-peakTree tool. It is unclear what the two algorithms can do individually, what are the overlapping capabilities, and what kind of benefit (either additional scientific information extracted or reduced effort to use) one can derive from the combination of them. Moreover, I think that the results derived in section 4.3 highlight some limitations of the radar Doppler analysis which I believe are worth a more extended discussion. I believe addressing these points does not require substantial work or additional experiments, but rather some work on the manuscript text. Thus, I recommend the paper to be published after minor revisions.*

**Response:** Thank you for taking the time to read our work and give detailed feedback. Your ideas on how the style of the paper and the presented methods can be improved were very helpful and we are addressing the points you raised below.

**Comment 3.2:** *Figure 1. I had some difficulties in using this figure to facilitate my understanding of the paper. First, as far as I understood, the training phase of PEAKO requires human intervention. In this case, I would add this to the input block of PEAKO. Further, PEAKO also identifies peaks itself, right? At least in subsequent figures there are PEAKO-found peaks which is not an output of PEAKO in fig.1 I would add the complete list of output of PEAKO to the figure, as well as the output that is not given to peakTree.*

**Response:** Thank you for addressing the comprehensibility of Fig. 1 and for raising the issue of PEAKO/ peakTree also being possibly used as standalone tools. This is in connection with your comment 3.4 and we will address the two comments together in the answer below.

**Comment 3.3:** *Section 3. Aren't the results derived from a very limited case study and using specific assumptions regarding particle properties (scattering and fallspeed)? A different case comprising different particles might be easier to manage (i.e. a mixture of ice, snow and graupel/hail) or much more difficult (small ice crystals mixed with supercooled liquid, or snow and drizzle). Also, the assumptions made in terms of scattering properties and fallspeed of particles might have a noticeable impact on the peak prominence and separation respectively. Is there a specific reason to use that particular case and simulation setup and derive general conclusions from that? I would suggest exploring some different options in terms of scattering and fallspeed to assess the effect of those assumptions on your conclusions, and perhaps to include a few cases (also as supplementary material) that might be completely fictitious, and not based on measurements, to illustrate some clearly easy separable situations and a very difficult one.*

**Response:** Thanks for sharing your thoughts about the PAMTRA-based peak separability study in Section 3, we greatly appreciate this valuable input!
We indeed did some sensitivity tests regarding fall speeds and scattering properties. As expected, the strongest changes occurred for particles with complex shape (especially the "ice_rimed" class). While this affects the number of peaks detected in that part of the spectrum, the detection of the liquid peak was barely affected.
As suggested, we decided to include a second, more simple, example in the appendix. Here, the forward-simulated PNSD is only bimodal with a stronger separation of the peaks and a liquid peak with higher reflectivity. For this situation, the liquid peak could be retrieved reliably also for higher turbulence broadening.
Together, these two studies provide some support on the constraints of (liquid) peak separability and help to better explain the PEAKO-peakTree results in the NASCENT case.

[Figure]

Figure 6: Fig. A1 (appendix): PEAKO-peakTree peak detection results in forward-simulated cloud radar Doppler spectra using PAMTRA a) number of detected peaks by PEAKO-peakTree vs. the number of spectral averages $n$ set in the forward simulation. Compared are two sets of peak finding parameters: fixed (indicated by open diamonds; a prominence threshold of $1.5\,\mathrm{dB}$, minimum width of $0.03\,\mathrm{m\,s^{-1}}$, and no smoothing along the velocity dimension) and trained individually for each $n$ (dots). The marker color represents the turbulence broadening in $\mathrm{m\,s^{-1}}$ set in the PAMTRA forward simulation. b) fraction of successfully detected liquid peaks in 100 forward-simulated spectra vs. turbulence broadening in $\mathrm{m\,s^{-1}}$ for radar settings with $n = 6, 10, 40, 80$ and 120. One example of a forward-simulated spectrum is shown as inset ($n = 120$ and $\sigma = 0.02\ \mathrm{m\,s^{-1}}$).

We are changing the concluding sentences of section 3 as follows:
"From our simulations, we can conclude that a) a minimum number of 20 - 40 spectral averages is desirable in radar settings optimized for peak detection, and that b) liquid peaks can only be separated for EDR values up to approximately $0.0002\,\mathrm{m^2\,s^{-3}}$ (translating to a $\sigma$ of approximately $0.06\,\mathrm{m\,s^{-1}}$) under such complex multi-peak situations. Appendix A shows that for simpler conditions (PNSDs featuring a much stronger liquid peak), a separation may however still be possible even for low values of $n$ and slightly higher values of turbulence broadening."
And we are adding the following to the discussion of Fig. 8:
"This can be explained by the findings in section 3 and Appendix A: If the number of spectral averages is low ($n < 20$), the liquid peak can only be separated by PEAKO-peakTree for low to moderate turbulence conditions, and if the liquid peak is very prominent in the Doppler spectrum (Fig. A1b)."

While the primary focus of section 3 was to explore the impact of turbulence broadening and number of averages for a given spectrum, we are considering a follow-up study into the direction of closure studies, especially as more comprehensive data sets become available (e.g. Henneberger et al., 2023)

**Comment 3.4:** *This is connected to my previous comment about Fig. 1. It is unclear when PEAKO and peakTree are used as separate instances and when they are combined. Fig. 1 and the discussion about PEAKO seem to suggest that the purpose of this algorithm is to facilitate the job of peakTree, though it is unclear what might have been different for peakTree without the use of PEAKO*

*and it seems that PEAKO is a valuable software by itself for peak-finding and it is also used with this purpose later in the results. The contribution of peakTree to the overall scientific analysis can also be emphasized because sometimes there is a feeling that PEAKO is used to identify peaks and peakTree is left without a clear purpose. This can be addressed in a number of ways:*

- *Restructure Fig. 1 as suggested in my previous comment including all input and output of the two software.*

- *Clearly write what is the added value of having peakTree ingesting the optimized configuration file of PEAKO. Or alternatively, state what one would have had to do with peakTree without the use of such optimized configuration.*

- *Declare at the beginning of subsections 4.2 and 4.3 what is the intended purpose of the use of PEAKO and peakTree. In particular what information is aimed to extract from the capabilities of peakTree to separate various peaks and their moments.*

**Response:** Thank you for pointing us to these inconsistencies / unclarities in our manuscript and especially for giving some hints on how to resolve them. This is connected to your first comment (comment 3.1). We restructured Figure 1 as follows: We show the inputs and outputs of both algorithms more clearly and also added the required action by a human expert labeller (😉) to the flowchart.

We have also added the following sentence to the beginning paragraph of section 2.2, where the two algorithms are introduced:

"While both of them can be used standalone, this work focuses on the added value of using PEAKO to derive the peak detection parameters which are then transferred to peakTree via a configuration file and applied for peak detection." Furthermore, at the end of section 2.2.1, we suggest to add the following: "PEAKO can be used as a standalone tool (Kalesse et al., 2019; Billault-Roux et al., 2023) to detect and save the indices of peaks in cloud radar Doppler spectrum files. However, for more in-depth analyses and fast processing of larger data sets, the use of peakTree is recommended."

In section 2.2.2, we suggest to rephrase the second paragraph as follows:

"One advantage of using peakTree instead of the PEAKO standalone peak detection functionality is that, for each node, the moments of the spectrum are calculated: The reflectivity Z, mean Doppler velocity MDV, spectrum width, skewness, and the linear depolarization ratio LDR for dual-polarization radars. Please note that, throughout this paper, negative MDV values represent downward velocities in the vertical column (i.e. towards the radar). Another advantage is the use of Numba (Lam et al., 2015), which means a considerable improvement in terms of computation time compared to PEAKO." and we also addeed the following sentence later in the text:

"It should be noted that peakTree can also be used without PEAKO's optimized peak detection parameters (Radenz et al., 2019; Ramelli et al., 2021). In this case, the peak detection parameters have to be chosen manually in a rather subjective try-and-error procedure. We thus suggest the combined use of PEAKO-peakTree."

Thank you also for the idea to describe the purpose of each case study at the beginning of the two sections containing the case studies from the DACAPO-PESO and the NASCENT campaigns. We agree that this aids the comprehensibility of the paper.

We added the following to the beginning of Section 4.2:

"In this case study, peakTree is applied to Doppler spectra data from the W-band and Ka-band cloud radars deployed during DACAPO-PESO using the optimized peak detection parameters yielded by PEAKO. The reflectivity of the liquid cloud peak is derived by peakTree using the thresholds defined in Eq. 1 and 2. The purpose of this comparison is twofold: Firstly, to verify that the peaks derived for the two individual radar systems match, which serves as a validation of the proposed PEAKO-peakTree methodology. Secondly, this case demonstrates the capability of PEAKO-peakTree to identify (supercooled) liquid water layers in mixed-phase clouds, which is of considerable value for various research applications. For the aforementioned comparison, we choose to analyze a case where multiple liquid-containing cloud layers were observed on 13 March 2019 in Punta Arenas. On this particular day..."

And the beginning of Section 4.3:

"We choose a second case study from the NASCENT data set to compare the cloud radar Doppler

spectrum peaks detected by peakTree to hydrometeor types observed in situ by HOLIMO. This case serves to illustrate the capabilities and limitations of the proposed PEAKO-peakTree technique for detecting and attributing peaks to different hydrometeor types in a complex situation. Furthermore, the effect of turbulence and the number of spectral averages, which was outlined in theory in Section 3, is observed and further explored in real radar measurements. For this purpose, we selected a study period on 12 November 2019 between 15:00 and 16:15 UTC, which has been previously described in detail by Pasquier et al. (2022)."

**Comment 3.5:** *Section 4.3 - The analysis of the results seems to describe a situation with a classic Arctic mixed-phase cloud consisting of ice particles with liquid at cloudtop and a more unusual additional liquid layer at the bottom. However, from the HOLIMO observations, the results of the forward simulations and the discussion at lines 480-493 suggest that this cloud, and especially its reflectivity signal, is largely dominated by supercooled drizzle, while the frozen mode is of little importance. Perhaps this is interesting and should be emphasized as a limitation of the spectral analysis tool because I have the feeling that without the HOLIMO instrument, one would have very rapidly attributed the main peak at $-1\,m\,s^{-1}$ to snow particles instead of supercooled drizzle while the faster falling emerging mode would have been classified as rimed snowflakes.*

**Response:** Thank you for sharing your thoughts on the supercooled drizzle peak. You are completely right, and the point you made should also be mentioned in the text. We added the following sentences to the discussion of "time 2" in Section 4.3:
"At this point, we would also like to point out that this particular case illustrates one general limitation of Doppler spectrum-based hydrometeor classification tools: the Doppler spectrum is largely dominated by supercooled drizzle, which constitutes the main peak at approx. $-1\,m\,s^{-1}$. Without the HOLIMO instrument, this peak would have likely been classified as snow particles rather than supercooled drizzle, while the faster-falling emerging mode would have been classified as rimed snowflakes rather than frozen drops."

**Comment 3.6:** *Fig. 7c it is surprising that the algorithm did not catch the supercooled drops peak which is quite obvious in the spectrum. Can you comment a bit on this and declare if you plan work on the algorithm in order to better catch these situations? I also have the feeling that a human would have labeled it and a pure ML algorithm such as PEAKO could have identified it as well. This is particularly discouraging because even the PAMTRA simulation seems to model the Doppler spectrum particularly well. Perhaps, one can envision using PAMTRA as a generator of synthetic Doppler spectra to train PEAKO on a very large number of situations without the need for a human intervention.*

**Response:** This comment kind of pinpoints the weakness of the proposed PEAKO-peakTree method. Noise peaks and liquid water peaks often have very similar properties, and the algorithm has a hard time detecting only (and all) hydrometeor peaks when the spectra are noisy. Due to the area-based similarity measure in PEAKO, there is a higher penalty when the algorithm detects wrong (spurious) peaks with high reflectivities than when it fails to detect narrow peaks with low reflectivity (e.g. liquid peaks). So, especially for liquid peaks, the algorithm has a weakness - however, the DACAPO-PESO case study (Section 4.2) has demonstrated that the liquid water peak detection can also work very well in other situations (for other radar settings, resulting in less noisy spectra). One should keep in mind here that the radar settings during the NASCENT experiment were really not optimized for Doppler spectrum peak detection, plus the conditions were very turbulent, and this is, so to say, "the acid test" for our algorithms.
(related: Question of another reviewer as to why we included these data at all in our study if there are so many issues with them: comment 4.14)
We have really racked our brains to come up with a better solution to better detect small peaks without getting lots of (wrong) noise peaks (cf. the discussion around Fig. 2 in comment 1.6), and are open for discussion on how the peak detection can be further improved. This might however involve changing the architecture of the peak detection in PEAKO and peakTree. One possible solution could be adding additional criteria to the peak detection function. E.g. a condition that two peaks cannot be closer than a certain Doppler velocity threshold, or they will be counted as one. We acknowledge

that our paper documents the current status of ongoing work, and is also aimed to spark scientific discussion about possible improvements to the method.

So, in short, to answer the question whether we are planning to work further on the algorithms to better catch these tricky situations, yes.

**Comment 3.7:** *Line 3 - The phrasing "convoluted by dynamical effects" puzzled me. It suggests that different hydrometeors would produce distinct peaks if not disturbed by turbulence or vertical and horizontal wind shear (combined with radar finite aperture and pulse duration). In reality, various particle spectra would overlap anyway. The caveat of addressing only particles with "sufficient different terminal fall velocities" seems again to not make it clear enough. Particles with clearly distinct peaks can still have significant tails in the velocity distribution and by a change in the total number concentration of one population the second one can appear in the spectrum or be completely masked by the tail of the other population. So, I would try to simplify this statement by clearly stating that Doppler spectra of different hydrometer types overlap and turbulence and other dynamical effects further smear the observed Doppler spectrum. But I understand that I am being very picky.*

**Response:** Thanks for the comment. We modified the Abstract and Introduction sections accordingly.

**Comment 3.8:** *Line 74, 177, 341, 349, 358, 366, 369, 370, 558 - the term "user" is a nice shortcut, but it feels odd. The person that is tasked to label peaks is not really a user. It might be a user of PEAKO, but again, it is not necessary, it could be anyone with sufficient "motivation" to perform such an effort. I would call this capable person: expert, labeller, decisor, identifier, and similar depending on the context. But again, I am very picky*

**Response:** You are right, the person labeling the peaks is not necessarily the user of PEAKO. We changed the wording to "(human) expert (labeller)" in the text.

**Comment 3.9:** *Section 3. I do not like the term "preconditions", or, at least, I do not understand these conditions come before what exactly. Why not just call them "conditions"?*

**Response:** thanks for noticing this, we changed the term as suggested to "conditions".

**Comment 3.10:** *I think the ice lollipops were first named ice lollies according to Keppas et al. 2017 ([https://agupubs.onlinelibrary.wiley.com/doi/full/10.1002/2017GL073441](https://agupubs.onlinelibrary.wiley.com/doi/full/10.1002/2017GL073441)). I know that Pasquier already called them ice lollipops, but it might be better to keep it consistent and avoid possible confusion in the future. Similarly at line 531, the term droplet lollipop seems misleading because, out of context it might be thought to be a strangely-shaped liquid drop.*

**Response:** Thanks for pointing this out! We agree and changed "lollipops" to "lollies" and also added the reference to the updated paper version.

**Comment 3.11:** *Fig. 6e In the label in the figure says number of peaks detected by peakTree in the caption PEAKO. As long as these are distinct software it is better to clarify which one is responsible for the peak detection.*

**Response:** Thanks for noticing. We changed the figure caption accordingly to "peakTree".

**Reviewer 4**

**Comment 4.1:** *This study describes updates to two separate algorithms – PEAKO and peakTree – and demonstrates their synergistic use through two separate case studies. The authors find that the combined algorithms are able to reliably identify multiple Doppler spectra peaks associated with distinct hydrometeor populations (e.g., supercooled liquid water and ice crystals) outside of cases of moderate-to-strong turbulence. The manuscript is generally well written, despite at times there being a lot of observation detail and acronyms to try to keep track of, and the figures are informative as well. I also appreciate the codes being provided to the community on Github. There are a number of things that I think would help improve the manuscript detailed below, but none of them rise to anything major scientifically. I therefore recommend this paper for publication after minor revisions.*

**Response:** We thank the reviewer for taking the time to carefully go through our manuscript and give detailed feedback.

**Comment 4.2:** *The main novelty of this paper seems to be the synergistic use of PEAKO and peakTree, which performs fairly well (at least for the case studies shown). However, on L161 it is described that PEAKO itself has a peak detection function that is used to find the optimal parameter settings compared to human-identified peaks. So in that way, couldn't PEAKO just be used to find and isolate the peaks itself without relying on peakTree? I suspect there are benefits to using peakTree and that it offers more capabilities (e.g., moment information rather than just the peak locations?), but this should be made a bit clearer to the reader up front who may similarly wonder why PEAKO isn't sufficient on its own.*

**Response:** We thank the reviewer for pointing us to this shortcoming of the manuscript. This is completely right and similar issues were raised by other referees, and we responded to them in detail in the answer to comment 3.4.

**Comment 4.3:** *I admit I might be missing something fundamental here, but it isn't clear to me how the peakTree species identification was decided. On L202, couldn't these parameters also correspond to cloud ice? What is it about these values that denotes only cloud liquid droplets? (This is seemingly agreed with on L394-395). Similarly, on L210, I feel like these criteria would also fit for moderate to relatively large liquid droplets, but here they are attributed only to frozen drops and rimed ice particles.*

**Response:** Yes, this is a very valid criticism and actually one of the main limitations of methods based on Doppler spectra in general. Regarding the identification of liquid cloud droplets, you are correct that these criteria can also be valid for small ice crystals. In section 2.2.2, we wanted to provide general thresholds that can be applied to all three radar systems used in our study, i.e. also the non-polarimetric JOYRAD94 radar. For dual-pol systems, an additional criterion can be applied that excludes peaks with elevated LDR, which is typical for columnar ice. Still, even if high-LDR peaks are excluded, one cannot be 100% sure that only liquid cloud droplets are selected by the defined rule. This also holds true for the VOODOO algorithm, which is based on Doppler spectra features, and a general limitation of Doppler spectrum-based approaches. To make this more clear, we added the following to section 2.2.2:
"It should be noted that liquid peaks in up- or downdrafts larger than $0.3\,\mathrm{m\,s^{-1}}$ are not detected by this simple selection rule and that small ice can result in peaks with similar properties. For dual-pol radars, an additional criterion can be added to filter peaks with elevated LDR, which are characteristic for columnar ice particles."
Large liquid droplets can to some extent be excluded from the analysis by applying the Cloudnet mask or another temperature-based selection criterion to only consider the ice and mixed-phase parts of the cloud and not the pixels classified as drizzle or rain. This, however, does not filter out supercooled drizzle, which was also noted by another reviewer (comment 3.5). It might be interesting to the reviewer to also read our response to the related issue raised there.
Furthermore, we modified the text about the fast-falling peak selection rule in Section 2.2.2 as follows:
"Furthermore, it is often of interest to cloud microphysical studies to consider hydrometeors with large fall velocities, which, in the temperature range for mixed-phase clouds between 0 and -40°C, usually correspond to large frozen drops and strongly rimed particles."

**Comment 4.4:** *More generally, I would be interested in some discussion/acknowledgement of what would happen to the performance of PEAKO-peakTree if non-Rayleigh scattering is present. Has this been investigated? Can it be included in training data? I know this is not the main motivation of the paper, but with Ka- and W-band radars this could reasonably be expected to be somewhat common.*

**Response:** Thank you for raising this important issue. Peaks due to Mie oscillations are mostly observed in rain, which we do not consider in this study. However, this is definitely something that should be mentioned in the paper. We added the following sentence to the introduction:
"Moreover, for large hydrometeors, Mie oscillation of the backscattered power can cause additional minima and maxima in the Doppler spectra, which is especially important at W-band (Tridon et al., 2017)"

**Comment 4.5:** *I would like to see a bit more objectivity regarding the comparison of VOODOO with the attenuated backscatter for the comparison on L389. While there is good agreement in general, there are a number of areas VOODOO picks up on that the lidar does not. Even when there is agreement, the VOODOO probabilities seem to be in much broader layers than the attenuation. Which is more realistic? Is it possible to overlay one field on the other for a more obvious comparison or do something quantitative beyond describing the agreement as "well" (L563)*

**Response:** We agree that comparing the fields in Fig. 4 is quite difficult and could be aided by an overlay plot. As suggested, we made an overlay plot of lidar attenuated backscatter and predicted probability for cloud droplet-containing pixels by VOODOO (Fig. 7).

This visualization corroborates our statement that there is good agreement between the two fields.

[Figure]

Figure 7: (a) VOODOO probability for liquid droplets (b) lidar attenuated backscatter (c) overlay plot of (a) and (b), the pink structure only contains $p_{cd}$ values > 0.4

But, as you noticed, in the higher cloud layers (beyond 3 km), VOODOO liquid-containing layers are noticeably thicker than those observed by the lidar. The layer of elevated lidar backscatter is located at the lower boundary of the pink structure representing the VOODOO liquid-containing layer in Fig. 7c. We believe it likely that the liquid-containing layers are indeed as thick as predicted

by VOODOO: The geometric extent of liquid water layers observed by the lidar is only meaningful up to the altitude where the signal is completely attenuated, underestimating the thickness of deep liquid-containing layers. VOODOO is based on Doppler spectra recorded by the cloud radar, which is able to penetrate thick cloud systems (Schimmel et al., 2022). The fact that the lidar sees the layer at the lower boundary of the liquid field detected by VOODOO indicates that the lidar signal gets completely attenuated at some point, but actually there is more liquid water above, which can be detected by VOODOO.

We suggest to include the overlay plot instead of the attenuated backscatter only in Fig. 4d in the paper. Furthermore, we suggest to add the following discussion to the text:

"Yellow-ish layers indicate a high probability for the presence of cloud droplets in the respective observation volume. For better comparability of the VOODOO-predicted $p_{CD}$ and the attenuated backscatter coefficient observed by the lidar, pixels with $p_{CD} > 0.65$ are overlaid on Fig. **??**d. In this visualization, it becomes apparent that these regions with elevated $p_{CD}$ match well with the areas exhibiting high attenuated backscatter coefficient in those parts of the cloud, which can be penetrated by the lidar. In higher cloud layers (beyond 3 km), VOODOO liquid-containing layers are noticeably thicker than those observed by the lidar, with the layer of elevated lidar backscatter being located at the lower boundary of the pink structure representing high $p_{CD}$ regions. This suggests that the geometric extent of liquid water layers is larger than observed by the lidar, because its signal becomes completely attenuated at some point in the cloud."

**Comment 4.6:** *L11: The use of "structure" here as a verb is a bit unclear to me. Can a different word or a more descriptive phrase be used to describe what is actually meant?*

**Response:** We changed the wording to "organize" instead of "structure".

**Comment 4.7:** *L173: For identified peaks with very different minima on either side of the peak (say, the leftmost peak out of many), which side is used to define the prominence? The side with the larger minimum value?*

**Response:** Always the side with the lower minimum of spectral reflectivity. If there is a minimum on the other side, the resulting peak will be split at a later step (this is the advantage of a recursive data structure).

**Comment 4.8:** *L179: Are the training data with peaks identified by humans done on data that has been smoothed at all, or is that data raw so that the impact of any smoothing/averaging can be evaluated?*

**Response:** The training data are not smoothed, the smoothing span and polynomial order and averaging window are defined by some of the PEAKO adjustable parameters, i.e. optimized during the training phase of PEAKO.

**Comment 4.9:** *L202: I might be missing something very fundamental, but it isn't clear to me how these simple criteria couldn't also refer to very small cloud ice particles, which should also have very low Z and terminal velocities. Is there a reason to believe any particles meeting this criteria should only correspond to liquid droplets?*

**Response:** Repetition of comment 4.3

**Comment 4.10:** *L217: If this output is being used to validate PEAKO-peakTree, can a bit more be said about VOODOO's accuracy or when it may fail?*

**Response:** We believe that this is actually already done sufficiently in Section 4.2, where we state: "When using VOODOO, one limitation needs to be considered however, i.e. that it is solely based on the cloud radar reflectivity Doppler spectra, leading to the potential misclassification of small and narrow spectral peaks caused by ice as liquid. In regions immediately beneath layers of SLW, ice

crystals newly formed through primary or secondary ice formation processes can be present. Due to their small diameters and consequently low fall velocities, these pristine ice crystals produce peaks in the cloud radar Doppler spectra that closely resemble those of liquid cloud droplets."
...
"As noted earlier, VOODOO can misclassify Doppler spectra with narrow ice peaks as liquid-containing."

**Comment 4.11:** *L226: I admit that I am not so familiar with this method of estimating EDR, but I would have assumed that EDR is more a function of spectrum width than MDV. At first glance it isn't clear to me why MDV itself would necessarily indicate anything about turbulence. Couldn't enhanced values of |MDV| just indicate updrafts and downdrafts that themselves don't have anything to do with turbulence, or does L232 indicate that it is binned 5-min values of MDV and their variability that are related to turbulence (so more like spectrum width of MDV values rather than the spectra itself)?*

**Response:** A similar issue was raised by another referee in comment 1.4, regarding the overall comprehensibility of Section 2.4. We have reorganized and rephrased the majority of this section and hope that it makes more sense now. You are right that the 5-minute values of MDV are used to calculate a power spectrum, which represents the variability of MDV in this time window.

**Comment 4.12:** *L251: Where did the value of 10 m-3 come from? How sensitive are the results to this value? I would have assumed interpolation would work better here. Depending on the number concentrations in the surrounding bins, couldn't this value still cause artificial peaks to be detected?*

**Response:** A very similar question was asked by another referee (comment 1.5) and is answered there.

**Comment 4.13:** *L310: Can the authors explain a bit more about why was this done?*

**Response:** line 310: "In addition, a liquid peak is only considered to be detected if its reflectivity and MDV deviate less than $1\,\mathrm{dB}$ and $0.1\,\mathrm{m\,s^{-1}}$ from the reference obtained for 150 coherent averages and weak turbulent broadening."
A criterion is needed to decide if the peak is correctly identified. Therefore, we use a reference obtained from a simulation with low turbulence and long averages, which produces a 'smooth spectrum' with only the liquid droplet population. The detected peaks are then compared to this reference in terms of the described criteria.

**Comment 4.14:** *L368: If there are this many issues with most of the JOYRAD94 chirps such that the training data is unreliable, why should this data even be incorporated into the study?*

**Response:** Good question. While we initially wanted to use the data to compare our Doppler spectra-based technique with balloon-borne in situ observations, it turned out along the way that they were also very well suited as kind of an "endurance test" for the methodology, testing the impacts of strong turbulence in combination with a low number of spectral averages.

**Comment 4.15:** *L382: It isn't clear to me how this alone is indicative of multiple embedded liquid water-containing layers. While maybe atypical of these clouds, couldn't this just signify layers of consistent updrafts that offset the MDV there?*

**Response:** True, such signatures in MDV alone are no evidence for the existence of liquid water layers. We changed the word "indicating" to "suggesting".

**Comment 4.16:** *L402: Given the disagreement between VOODOO/peakTree and the lidar attenuation, is there a reason to believe VOODOO/peakTree over the attenuation? Attenuation in the presence of liquid water seems definitive and I don't know why there wouldn't be any in this region.*

**Response:** line 402: "Compared to lidar-detected liquid layers, further layers near the cloud tops with temperatures of around -20°C... and -12°C, pointing to the existence of SLW-containing layers, are evident in VOODOO and PEAKO-peakTree based results."
This is the region where the lidar signal is already completely attenuated by the liquid layers below, meaning that there can't be any more signal in the attenuated backscatter above. Both VOODOO and PEAKO-peakTree find additional liquid layers there. The peaks identified as "liquid" by peakTree have very low LDR, which is another indication towards liquid (and not small ice). See also our responses to comment 4.5 and comment 4.17.

**Comment 4.17:** *I find the use of LDR really compelling in this context, as it helps address the point above. I highly suggest the authors include a plot of LDR in the manuscript for readers. More generally, could LDR not be incorporated as part of the training data set (when available)?*

**Response:** Fig.8 shows the LDR of the liquid node identified by the peakTree selection rule. We

[Figure]

Figure 8: LDR (measured by MIRA-35) of the liquid peak for the case study from 13 March 2019, Punta Arenas. In the lowest range gates, there are artificially high values, which should be ignored.

state in the paper:
"For the liquid peaks identified in the MIRA-35 spectra by the PEAKO-peakTree toolkit in Fig. 4f, the LDR consistently ranges below -25 dB (not shown), which is a strong indication that those peaks are indeed being caused by liquid water."
We believe that this plot does not need to be included in the paper, as it does not show any interesting features.

**Comment 4.18:** *L451: When I first read "correlation" I assumed in meant positive correlation. It may help readers to clarify "inverse correlation" as it is being introduced.*

**Response:** We thank the reviewer for pointing us to this inconsistency. This was also noticed by another referee in comment 1.8 and addressed in our response. We propose to replace the word "correlation" with "link":
"Upon initial review of Fig 6c and e, a link between the number of detected peaks and the EDR becomes apparent."

**Comment 4.19:** *Figure 7: What explains the difference in the Doppler velocity associated with the peak between the observed spectra and the simulated? (E.g., 7a shows a peak at around -0.3 m s$^{-1}$ while the simulated peak is near -1.5 m s$^{-1}$ due to drizzle).*

**Response:** This difference can be explained by vertical air motion, in this case an updraft in the observed spectrum, causing the spectrum to extend up to velocities of $+1\,\mathrm{m\,s^{-1}}$. In our PAMTRA simulations, vertical air motion is set to $0\,\mathrm{m\,s^{-1}}$ (this is stated in Section 2.5 of our manuscript). We

think that this difference should be briefly discussed in the text and added the following to Section 4.2:
"It should be noted that the vertical air velocity is set to $0\,\mathrm{m\,s^{-1}}$ in the PAMTRA simulations, while the observed spectra can be impacted by vertical air motions, moving them along the Doppler velocity axis."

**Comment 4.20:** *L575: What is Cloudnet? Unless I missed it, this is the first mention of it that I have seen.*

**Response:** We thank the reviewer for noticing this! We did mention the Cloudnet target classification in section 2.2, but never introduced Cloudnet itself. In the introduction, ACTRIS is mentioned, into which Cloudnet has been incorporated recently. We propose to add "within Cloudnet" after ACTRIS is mentioned in the introduction to make this clearer:
"observations need to be available, such as those provided by the platforms of the Aerosol, Clouds and Trace Gases Research Infrastructure (ACTRIS, https://www.actris.eu) within Cloudnet, or the Atmospheric Radiation Measurement (ARM, https://www.arm.gov/) observations."

**Comment 4.21:** *L580: Would RhoHV not also be useful in this capacity?*

**Response:** You are right, and incorporating this into peakTree is actually work in progress. For STSR radars, like the LIMRAD94, this variable would be even more direct than using the LDR of sub-peaks.

**Comment 4.22:** *Typos/Grammar/Etc.*
*1. L36: "clouds" should be "cloud" at the end of this line.*
*2. L141: "was" should be "were"*
*3. L165: It would be clearer if "has meanwhile" was changed to "have now"*
*4. L271: There is an extra comma after 0.2.*
*5. L524: "of" should cone after "existence"*
*6. L580: Should "width" be "spectrum width"?*

**Response:** Thanks for spotting all those. We have made the suggested changes.

**References**

[revised manuscript text omitted]